# Transcripts with high distal heritability mediate genetic effects on complex metabolic traits

Anna L. Tyler [1,5], J. Matthew Mahoney [1,5], Mark P. Keller [2], Candice N. Baker [1], Margaret Gaca[1], Anuj Srivastava[3], Isabela Gerdes Gyuricza[1], Madeleine J. Braun[1], Nadia A. Rosenthal [1,4], Alan D. Attie [2], Gary A. Churchill [1] ✉ & Gregory W. Carter [1] ✉

Although many genes are subject to local regulation, recent evidence suggests that complex distal regulation may be more important in mediating phenotypic variability. To assess the role of distal gene regulation in complex traits, we combine multi-tissue transcriptomes with physiological outcomes to model diet-induced obesity and metabolic disease in a population of Diversity Outbred mice. Using a novel high-dimensional mediation analysis, we identify a composite transcriptome signature that summarizes genetic effects on gene expression and explains 30% of the variation across all metabolic traits. The signature is heritable, interpretable in biological terms, and predicts obesity status from gene expression in an independently derived mouse cohort and multiple human studies. Transcripts contributing most strongly to this composite mediator frequently have complex, distal regulation distributed throughout the genome. These results suggest that trait-relevant variation in transcription is largely distally regulated, but is nonetheless identifiable, interpretable, and translatable across species.

Evidence from genome-wide association studies (GWAS) suggests that most heritable variation in complex traits is mediated through regulation of gene expression. The majority of trait-associated variants lie in gene regulatory regions[1–7], suggesting a relatively simple causal model in which a variant alters the homeostatic expression level of a nearby (local) gene which, in turn, alters a trait. Statistical methods such as transcriptome-wide association studies (TWAS)[8–11] and summary data-based Mendelian randomization (SMR)[10] have used this idea to identify genes associated with multiple disease traits[12–15]. However, despite the great promise of these methods, explaining trait effects with local gene regulation has been more difficult than initially assumed[16,17]. Although trait-associated variants typically lie in non-coding, regulatory regions, these variants often have no detectable effects on gene expression[16] and tend not to co-localize with expression quantitative trait loci (eQTLs)[17,18].

These observations suggest that the relationship among genetic variants, gene expression, and organism-level traits is more complex than the simple, local model.

In recent years the conversation around the genetic architecture of common disease traits has been addressing this complexity, and there is increased interest in more distant (distal) genetic effects as potential drivers of trait variation[15,18–21]. In general, distal effects are defined as being greater than 4 or 5Mb away from the transcription start site of a given gene. We use the terms local and distal rather than *cis* and *trans* because *cis* and *trans* have specific biochemical meanings[22], whereas local and distal are defined only by genomic position. The importance of distal genetic effects is proposed in the omnigenic model, which posits that trait-driving genes are cumulatively influenced by many distal variants. In this view, the heritable

[1]The Jackson Laboratory, Bar Harbor, Maine, USA. [2]University of Wisconsin-Madison, Biochemistry Department, Madison, WI, USA. [3]The Jackson Laboratory for Genomic Medicine, Farmington, CT, USA. [4]National Heart and Lung Institute, Imperial College, London, UK. [5]These authors contributed equally: Anna L. Tyler, J. Matthew Mahoney. ✉e-mail: gary.churchill@jax.org; gregory.carter@jax.org

transcriptomic signatures driving clinical traits are an emergent state arising from the myriad molecular interactions defining and constraining gene expression. Consistent with this view, it has been suggested that part of the difficulty in explaining trait variation through local eQTLs may arise in part because gene expression is not measured in the appropriate cell types[16], or cell states[23], and thus local eQTLs influencing traits cannot be detected in bulk tissue samples. This context dependence emphasizes the essential role of complex regulatory and tissue networks in mediating variant effects. The mechanistic dissection of complex traits in this model is more challenging because it requires addressing network-mediated effects that are weaker and greater in number. However, the comparative importance of distal effects over local effects is currently only conjectured and challenging to address in human populations.

To assess the role of wide-spread distal gene regulation in the genetic architecture of complex traits, we used genetically diverse mice as a model system. In mice we can obtain simultaneous measurements of the genome, transcriptome, and phenome in all individuals. We used diet-induced obesity and metabolic disease as an archetypal example of a complex trait. In humans, these phenotypes are genetically complex with hundreds of variants mapped through GWAS[24,25] that are known to act through multiple tissues[26,27]. Likewise in mice, metabolic traits are also genetically complex[28] and synteny analysis implicates a high degree of concordance in the genetic architecture between species[12,28]. Furthermore, in contrast to humans, in mice we have access to multiple disease-relevant tissues in the same individuals with sufficient numbers for adequate statistical power.

We generated two complementary data sets: a discovery data set in a large population of Diversity Outbred (DO) mice[29], and an independent validation data set derived by crossing inbred strains from the Collaborative Cross (CC) recombinant inbred lines[30] to form CC recombinant inbred intercross (CC-RIX) mice. Animals in populations were maintained on a high-fat, high-sugar diet to model diet-induced obesity and metabolic disease[12].

The DO population and CC-RIX were derived from the same eight inbred founder strains: five classical lab strains and three strains more recently derived from wild mice[29], representing three subspecies and capturing 90% of the known variation in laboratory mice[31]. The DO mice are maintained with a breeding scheme that ensures equal contributions from each founder across the genome, thus rendering almost the whole genome visible to genetic inquiry and maximizing power to detect eQTLs[29]. The CC mice were initially intercrossed to recombine the genomes from all eight founders, and then inbred for at least 20 generations to create recombinant inbred lines[30–32]. Because these two populations have common ancestral haplotypes but highly distinct kinship structure, we could directly and unambiguously compare the local genetic effects on gene expression at the whole-transcriptome level while varying the population structure driving distal regulation.

In the DO population, we paired clinically relevant metabolic traits, including body weight and plasma levels of insulin, glucose and lipids[12], with transcriptome-wide gene expression in four tissues related to metabolic disease: visceral adipose tissue (gonadal fat pad), pancreatic islets, liver, and skeletal muscle. We measured similar metabolic traits in a CC-RIX population and gene expression from three of the four tissues used in the DO: visceral adipose tissue (gonadal fat pad), liver, and skeletal muscle. Measuring gene expression in multiple tissues is critical to adequately assess the extent to which local gene regulation varies across the tissues and whether such variability might account for previous failed attempts to identify trait-relevant local eQTLs. Because the CC-RIX carry the same founder alleles as the DO, the local gene regulation is expected to match between the populations. However, because the alleles are recombined throughout the genome, distal effects are expected to vary from those in the DO, allowing us to directly assess the role of distal gene regulation in driving trait-associated transcript variation. To mechanistically dissect distal effects on metabolic disease, we developed a novel dimension reduction framework called high-dimensional mediation analysis (HDMA) to identify the heritable transcriptomic signatures driving trait variation, which we compared between mouse populations and to human data sets with measured adipose gene expression. Together, these data enable a comprehensive view into the genetic architecture of metabolic disease.

## Results

### Genetic variation contributed to wide phenotypic variation
Although the environment was consistent across the DO mice, the genetic diversity present in this population resulted in widely varying distributions across physiological measurements (Fig. 1). For example, body weights of adult individuals varied from less than the average adult C57BL/6J (B6) body weight to several times the body weight of a B6 adult in both sexes (Males: 18.5–69.1g, Females: 16.0–54.8g) (Fig. 1A). Fasting blood glucose (FBG) also varied considerably (Fig. 1B), although few of the animals had FBG levels that would indicate pre-diabetes (19 animals, 3.8%), or diabetes (7 animals, 1.4%) according to previously developed cutoffs (pre-diabetes: FBG ≥250 mg/dL, diabetes: FBG ≥300, mg/dL)[33]. Males had higher FBG than females on average (Fig. 1C) as has been observed before suggesting either that males were more susceptible to metabolic disease on the high-fat, high-sugar (HFHS) diet, or that males and females may require different FBG thresholds for pre-diabetes and diabetes.

Body weight was strongly positively correlated with food consumption (Fig. 1D; Linear regression $R^2 = 0.51$; beta coefficient $= 12.6 \pm 0.57$ standard error; $t = 22.2$; $p < 2.2^{-16}$) and FBG (Fig. 1E; Linear regression $R^2 = 0.21$; beta coefficient $= 2.49 \pm 0.22$ standard error; $t = 11.34$; $p < 2.2^{-16}$) suggesting a link between behavioral factors and metabolic disease. However, the heritability of this trait and others (Fig. 1F) indicates that genetics contribute substantially to correlates of metabolic disease in this population.

The trait correlations (Fig. 1G) showed that most of the metabolic trait pairs were only modestly correlated, which, in conjunction with the trait decomposition (Supplementary Fig. 1), suggests complex relationships among the measured traits and a broad sampling of multiple heritable aspects of metabolic disease including overall body weight, glucose homeostasis, and pancreatic function.

### Distal heritability correlated with phenotype relevance
It is widely assumed that variation in traits is mediated through local regulation of gene expression. To test this assumption, we measured transcriptome-wide gene expression in four tissues–adipose, liver, pancreatic islet, and skeletal muscle–in the DO cohort. (Basic results from a standard eQTL analysis[34] (Methods) are available in Supplementary Fig. 2). We estimated the local genetic contribution to each transcript as the variance explained by the haplotype probabilities at the genetic marker closest to the gene transcription start site. We estimated the distal heritability as the heritability of the residuals after local haplotype had been accounted for (Methods). Importantly, this estimate was not based on distal eQTL, but rather the unlocalized contribution of the genome after removing the local genetic effect.

Overall, local and distal genetic factors contributed approximately equally to transcript abundance. In all tissues, both local and distal factors explained between 8% and 17% of the variance in the median transcript (Fig. 2A). This 50% contribution of local genetic variation to transcript abundance contrasts with findings in humans in which local variants have been found to explain only 20–30% of total heritability, while distal effects explain the remaining 70–80%[35,36]. This discrepancy may arise due to the high degree of linkage disequilibrium in the DO mice compared to human populations and to the high degree of confidence with which we can estimate ancestral haplotypes in this population. At each position in the mice we can estimate ancestral haplotype with a high degree of accuracy. Haplotype at any given genetic marker captures genomic information from a relatively large genomic region

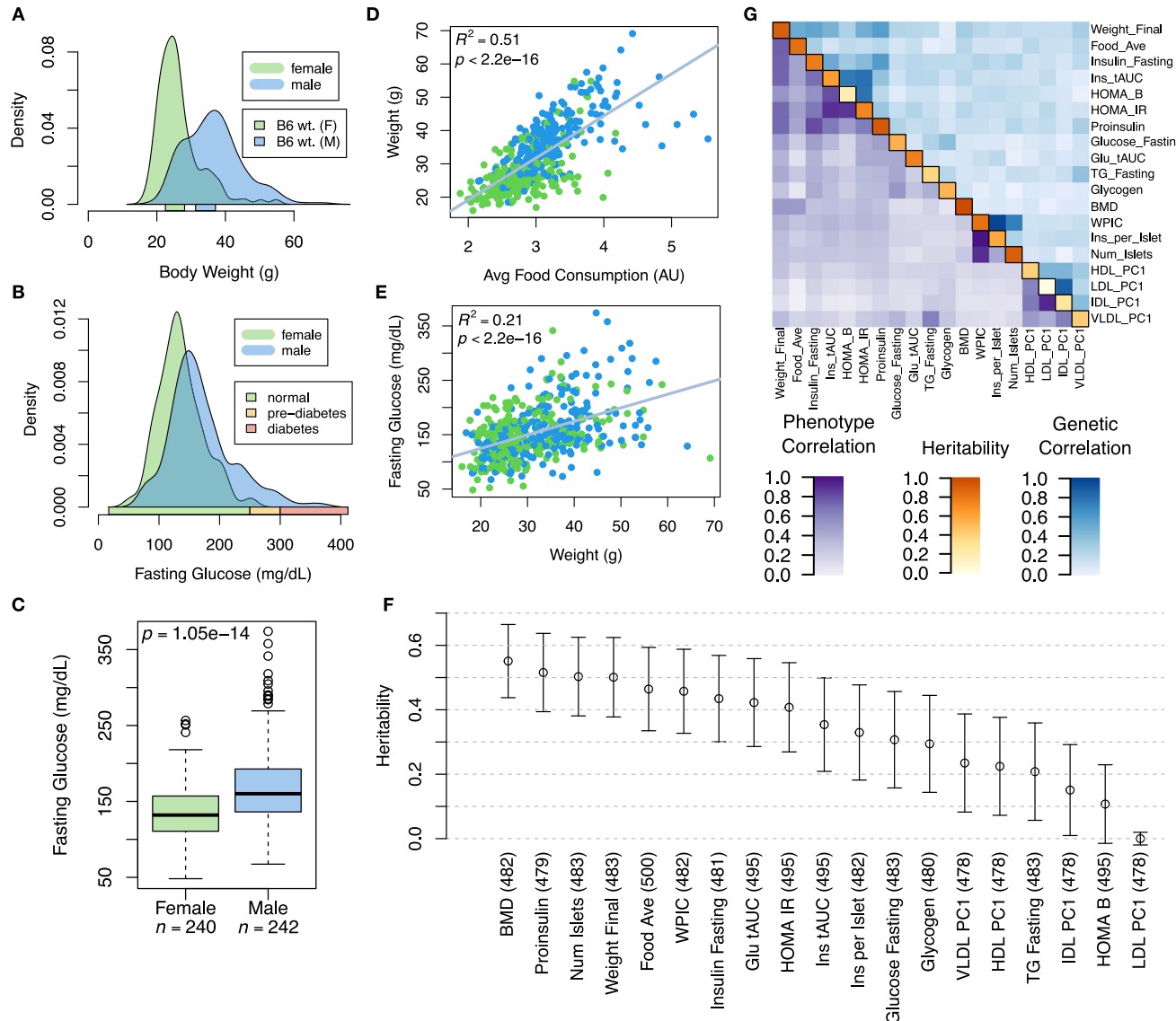

**Fig. 1 | Clinical overview. A** Distributions of final body weight in female (green) and male (blue) diversity outbred mice. The average B6 female and male adult weights at 24 weeks of age are indicated by green and blue bars respectively on the *x*-axis. **B** Distributions of fasting glucose in female (green) and male (blue) DO mice. Normal (green), pre-diabetic (yellow), and diabetic (red) fasting glucose ranges for mice are shown by colored bars along the *x*-axis. **C** Males (*n* = 242) had higher fasting blood glucose on average (mean = 170.0 mg/dL) than females (green *n* = 240, mean = 136) (two-sided Welch's t test: *t* = 8.02, df = 428.9, 95% CI of difference = 25.4 to 42.0 mg/dL; *p* = 1.05 × 10⁻¹⁴). Lines in boxes correspond to the median; lower and upper edges of boxes indicate the first and third quartiles; whiskers indicate the first and third quartiles ± 1.5 times the interquartile range; dots indicate outliers beyond 1.5 times the interquartile range. **D** The relationship between food consumption and body weight for female (green) and male (blue) DO mice. (Linear regression $R^2$ = 0.51; beta coefficient = 12.6 ± 0.57 standard error; *t* = 22.2; *p* < 2. 2⁻¹⁶). **E** Relationship between body weight and fasting glucose for

female (green) and male (blue) DO mice. (Linear regression $R^2$ = 0.21; beta coefficient = 2.49 ± 0.22 standard error; *t* = 11.34; *p* < 2. 2⁻¹⁶). In D and E, blue lines show line of best fit. **F** Data presented are heritability estimates for each physiological trait. Bars show standard error of each estimate. The number of animals used in each estimate is shown in parentheses after each trait name. **G** Correlation structure between pairs of physiological traits. The upper and lower triangle show the Pearson correlation coefficients (*r*) between LOD traces of trait pairs (blue) and trait pairs (purple) respectively. The diagonal (orange) shows the estimated heritability of each trait. BMD - bone mineral density, WPIC - whole pancreas insulin content, Glu tAUC - glucose total area under the curve, HOMA IR - homeostatic measurement of insulin resistance, HOMA B - homeostatic measure of beta cell health, VLDL - very low-density lipoprotein, LDL - low-density lipoprotein, IDL - intermediate density lipoprotein, HDL - high-density lipoprotein, TG - triglyceride. Source data are provided as a Source Data file.

surrounding each marker. In contrast, there is a much higher degree of recombination in human populations and ancestral haplotypes are more numerous and more difficult to estimate than in the mice. Thus in the mice, each marker may capture more local regulatory variation than SNPs or estimated haplotypes capture in humans. It has been found that transcripts with mulitple local eQTL have higher local heritability than transcripts with single local eQTL[37]. Because of the high diversity in the DO and the high rates of linkage disequilibrium, it is possible that there

are more local variants regulating transcription creating a proportionally larger effect of local regulation.

To assess the importance of genetic regulation of transcript levels to clinical traits, we compared the local and distal heritabilities of transcripts to their trait relevance. We defined trait relevance for a transcript as its maximum absolute Spearman correlation coefficient (*ρ*) across all traits (Methods). The local heritability of transcripts was negatively associated with their trait relevance (Fig. 2B), suggesting

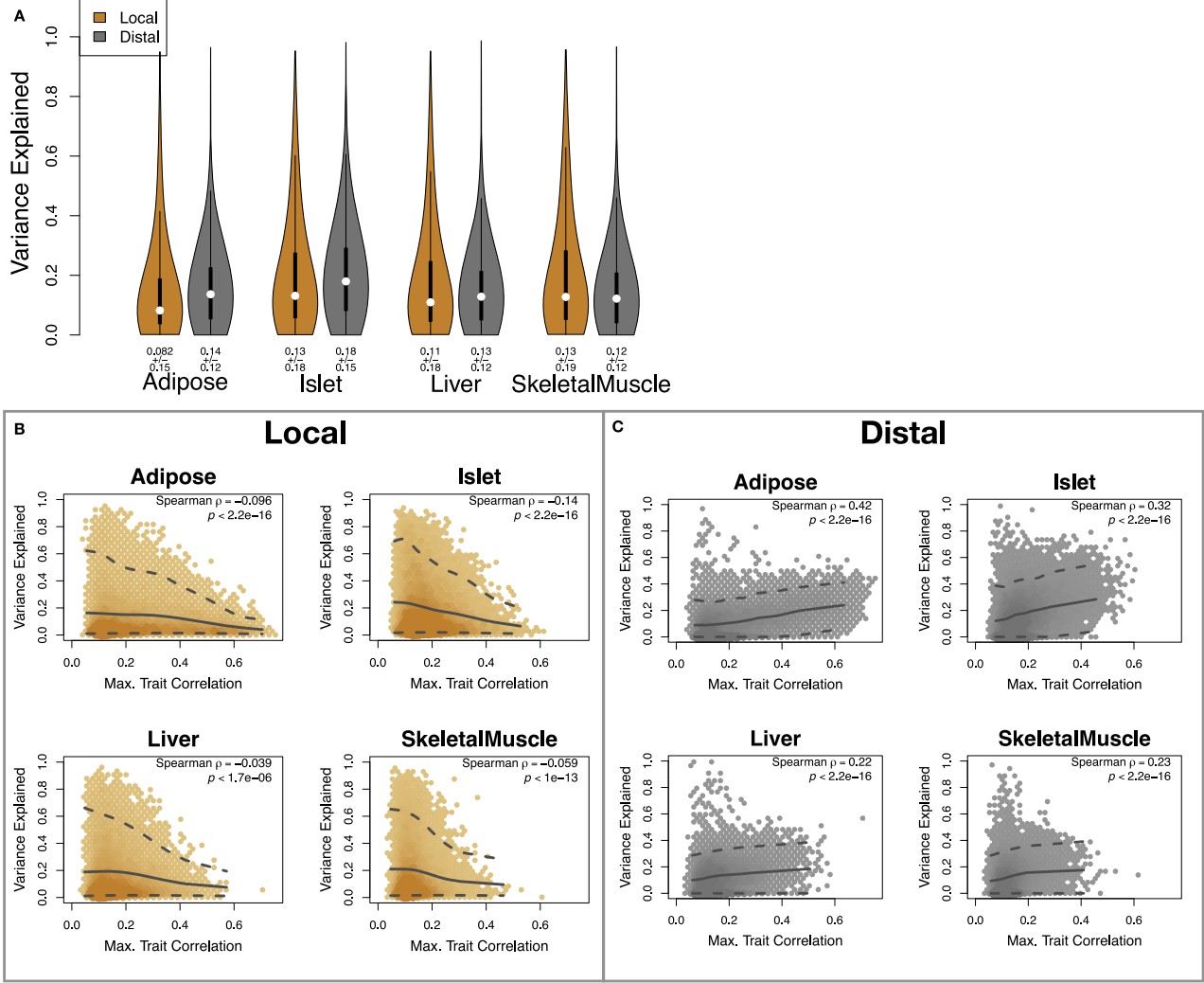

**Fig. 2 | Transcript heritability and trait relevance. A** Distributions of local (brown) and distal (gray) heritability of transcripts across the four tissues. Overall local and distal factors contribute equally to transcript heritability. Each distribution contains 14102 transcripts. Numbers below distributions indicate the median and standard deviation of each. **B** local (brown) and (**C**) distal (gray) heritability and trait relevance across all four tissues. Here trait relevance is defined as the maximum correlation between the transcript and all traits. The upper and lower dashed line in each panel show the 95th and 5th percentile correlation respectively. The solid line shows the mean trait correlation in transcripts with increasing variance explained either locally (**B**) or distally (**C**). Transcripts that are highly correlated with traits tend to have low local heritability and high distal heritability. All *p* values from Spearman rank correlation tests are two-sided. No adjustments were made for multiple comparisons. Source data are provided as a Source Data file.

that the more local genotype influenced transcript abundance, the less effect this variation had on the measured traits. Conversely, the distal heritability of transcripts was positively associated with trait relevance (Fig. 2C). That is, transcripts that were more highly correlated with the measured traits tended to be distally, rather than locally, heritable. This pattern was consistent across all tissues. This finding is also consistent with previous observations that transcripts with low local heritability explain more expression-mediated disease heritability than transcripts with high local heritability[19]. However, the positive relationship between trait correlation and distal heritability demonstrated further that there are diffuse genetic effects throughout the genome converging on trait-related transcripts.

## High-Dimensional Mediation Analysis identified a high-heritability composite trait that was mediated by a composite transcript

The above univariate analyses establish the importance of distal heritability for trait-relevant transcripts. However, the number of

transcripts dramatically exceeds the number of phenotypes. Thus, we expect the heritable, trait-relevant transcripts to be highly correlated and organized according to coherent, biological processes representing the mediating endophenotypes driving clinical trait variation. To identify these endophenotypes in a theoretically principled way, we developed a novel dimension-reduction technique, high-dimension mediation analysis (HDMA), that uses the theory of causal graphical models to identify a transcriptomic signature that is simultaneously 1) highly heritable, 2) strongly correlated to the measured phenotypes, and 3) conforms to the causal mediation hypothesis (Fig. 3). In HDMA, we first use a linear mapping called kernelization to dimension-reduce the genome, transcriptome, and phenome to kernel matrices $G_K$, $T_K$ and $P_K$ respectively, which each have the dimensions $n \times n$ where $n$ is the number of individuals (Methods). These kernel matrices describe the relationships among the individual mice in genome space, transcriptome space, and phenome space and ensure that these three omic spaces have the same dimensions, and thus the same weight in the analysis. If not dimension-reduced, the transcriptome would outweigh

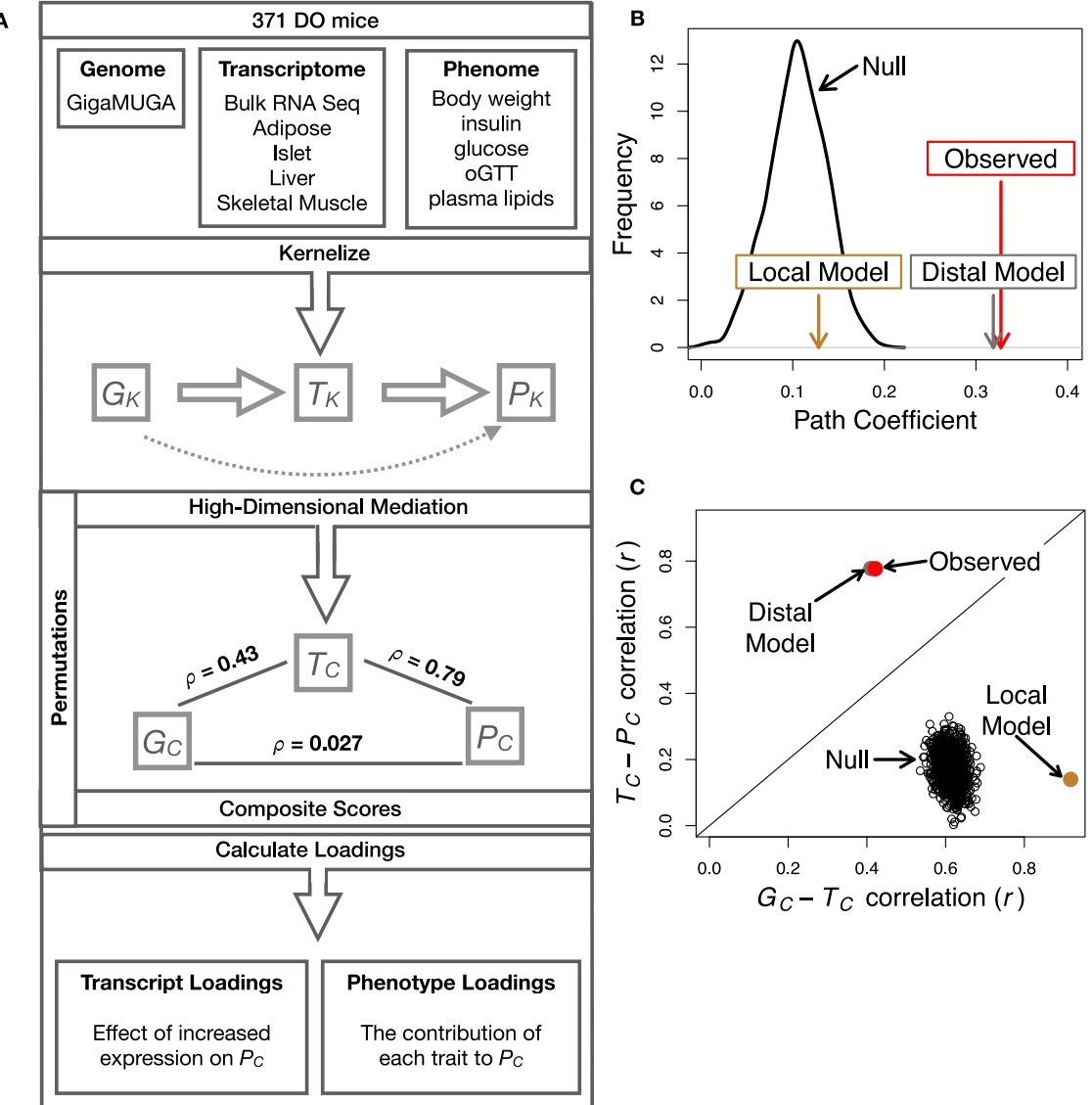

**Fig. 3 | High-dimensional mediation. A** Workflow indicating major steps of high-dimensional mediation. The genotype, transcriptome, and phenotype matrices were kernelized to yield single matrices representing the relationships between all individuals for each data modality ($G_K$ = genome kernel, $T_K$ = transcriptome kernel; $P_K$ = phenome kernel). High-dimensional mediation was applied to these matrices to maximize the direct path $G \to T \to P$, the mediating pathway (arrows), while simultaneously minimizing the direct $G \to P$ pathway (dotted line). The composite vectors that resulted from high-dimensional mediation were $G_C$, $T_C$, and $P_C$. The partial correlations $\rho$ between these vectors indicated perfect mediation. Transcript and trait loadings were calculated as described in the methods. **B** The null distribution of the path coefficient derived from 10,000 permutations. Comparisons are shown to the observed path coefficient (red) the path coefficient using a distal-only model (gray) and the path coefficient using the local-only model (brown). **C.** The null distribution of the $G_C$-$T_C$ correlation vs. the $T_C$-$P_C$ correlation. Comparisons are shown to the observed values (red), and those derived from the distal-only model (gray) and the local-only model (brown). Source data are provided as a Source Data file.

the phenome in the model. We then projected these $n \times n$-dimensional kernel matrices onto one-dimensional scores–a composite genome score ($G_C$), a composite transcriptome score ($T_C$), and a composite phenome score ($P_C$)–and used the univariate theory of mediation to constrain these projections to satisfy the hypotheses of perfect mediation, namely that upon controlling for the transcriptomic score, the genome score is uncorrelated to the phenome score. A complete mathematical derivation and implementation details for HDMA are available in the Methods.

Using HDMA we identifed the major axis of variation in the transcriptome that was consistent with mediating the effects of the genome on metabolic traits (Fig. 3). Figure 3A shows the partial correlations ($\rho$) between the pairs of these composite vectors. The partial correlation between $G_C$ and $T_C$ was 0.43, and the partial

correlation between $T_C$ and $P_C$ was 0.79. However, when the transcriptome was taken into account, the partial correlation between $G_C$ and $P_C$ was effectively zero (0.027). $P_C$ captured 30% of the overall trait variance, and its estimated heritability was 0.71 $\pm$ 0.084, which was higher than any of the measured traits (Fig. 1F). Thus, HDMA identified a maximally heritable metabolic composite trait and a highly heritable component of the transcriptome that are correlated as expected in the perfect mediation model.

As discussed in the Methods, HDMA is related to a generalized form of canonical correlation analysis (CCA). Standard CCA is prone to over-fitting because in any two large matrices it can be trivial to identify highly correlated composite vectors[38]. To assess whether our implementation of HDMA was similarly prone to over-fitting in a high-dimensional space, we performed permutation testing. We permuted

the individual labels on the transcriptome matrix 10,000 times and recalculated the path coefficient, which is the correlation of $G_C$ and $T_C$ multiplied by the correlation of $T_C$ and $P_C$. This represents the strength of the path from $G_C$ to $P_C$ that is putatively mediated through $T_C$. The permutations preserved the correlation between the genome and phenome, but broke the correlations between the genome and the transcriptome, as well as between the transcriptome and the phenome. We could thus test whether, given a random transcriptome, HDMA would overfit and identify apparently mediating transcriptomic signatures in random data. The null distribution of the path coefficient is shown in Fig. 3B, and the observed path coefficient from the original data is indicated by a red arrow. The observed path coefficient was well outside the null distribution generated by permutations (empirical $p < 10^{-16}$). Figure 3C illustrates this observation in more detail. Although we identified high correlations between $G_C$ and $T_C$, and modest correlations between $T_C$ and $P_C$ in the null data (Fig. 3C), these two values could not be maximized simultaneously in the null data. In contrast, the red dot shows that in the real data both the $G_C$-$T_C$ correlation and the $T_C$-$P_C$ correlation could be maximized simultaneously suggesting that the path from genotype to phenotype through the transcriptome is highly non-trivial and is identifiable in this case.

To test whether the presence of local eQTLs affected the result, we generated two additional transcriptomic kernel matrices. We generated a local-effects only kernel using only locally determined gene expression and a distal-effects only kernel using only distally determined gene expression, i.e., the effects of local haplotype were regressed out. The path coefficient identified using the local kernel was not significantly different from the null (Fig. 3B), suggesting that locally determined gene expression does not mediate the effects of the genome on the phenome. In contrast, the path coefficient identified using the distal kernel, was highly significant and indistinguishable from that identified using the full transcriptome.

Further, the $G_C$-$T_C$ and $T_C$-$P_C$ correlations derived from the distal kernal were indistinguishable from those derived from the original transcriptomic kernel. In contrast, the $G_C$-$T_C$ correlation derived with the local kernel was high (Pearson correlation $r = 0.92$), reflecting the fact that the local transcriptomic kernel was derived directly from local haplotypes. The $T_C$-$P_C$ correlation, however, was low (Pearson correlation $r = 0.14$), suggesting that the these locally derived transcripts were not highly related to phenotype. In other words, mice that shared many local eQTL were not highly similar in trait space. Taken together, these results suggest that composite vectors derived from the measured transcriptomic kernel represent genetically determined variation in phenotype that is mediated through genetically determined variation in transcription, and that this genetically deremined variation in transcription is largely driven by distal factors.

## Body weight and insulin resistance were highly represented in the expression-mediated composite trait

Each composite score is a weighted combination of the measured variables. The magnitude and sign of the weights, called loadings, correspond to the relative importance and directionality of each variable in the composite score. The loadings of each measured trait onto $P_C$ indicate how much each contributed to the composite phenotype. Body weight (Weight_Final) contributed the most (Fig. 4A), followed by homeostatic insulin resistance (HOMA_IR) and fasting plasma insulin levels (Insulin_Fasting). We can thus interpret $P_C$ as an index of metabolic disease (Fig. 4B). Individuals with high values of $P_C$ had a higher metabolic disease index (MDI) and greater metabolic disease, including higher body weight and higher insulin resistance. We refer to $P_C$ as MDI going forward. Traits contributing the least to MDI were measures of cholesterol and pancreas composition. Thus, when we interpret the transcriptomic signature identified by HDMA, we are explaining primarily the putative transcriptional mediation of body weight and insulin resistance, as opposed to cholesterol measurements.

## High-loading transcripts had low local heritability, high distal heritability, and were linked mechanistically to obesity

We interpreted large loadings onto transcripts as indicating strong mediation of the effect of genetics on MDI. Large positive loadings indicate that higher expression was associated with a higher MDI (i.e., higher risk of obesity and metabolic disease on the HFHS diet) (Fig. 4C–E). Conversely, large negative loadings indicate that high expression of these transcripts was associated with a lower MDI (i.e., lower risk of obesity and metabolic disease on the HFHS diet) (Fig. 4C–E). Figure 4D compares the observed transcript loading distributions to null distributions and indicates how many transcripts in each tissue had large positive and negative loadings. A direct comparison of the tissues can be seen in Supplementary Fig. 3. We used gene set enrichment analysis (GSEA)[39,40] to look for biological processes and pathways that were enriched at the top and bottom of this list (Methods).

In adipose tissue, both GO processes and KEGG pathway enrichments pointed to an axis of inflammation and metabolism (Supplementary Figs. 4 and 5). GO terms and KEGG pathways associated with inflammation were positively associated with MDI, indicating that increased expression in inflammatory pathways was associated with a higher burden of disease. It is well established that adipose tissue in obese individuals is inflamed and infiltrated by macrophages[41–45], and the results here suggest that this may be a dominant heritable component of metabolic disease.

The strongest negative enrichments in adipose tissue were related to mitochondial activity in general, and thermogenesis in particular (Supplementary Figs. 4 and 5). Genes in the KEGG oxidative phosphorylation pathway were almost universally negatively loaded in adipose tissue, suggesting that increased expression of these genes was associated with reduced MDI (Supplementary Fig. 6). Consistent with this observation, it has been shown previously that mouse strains with greater thermogenic potential are also less susceptible to obesity on an obesigenic diet[46].

Transcripts associated with the citric acid cycle as well as the catabolism of the branched-chain amino acids (valine, leuceine, and isoleucine) were strongly enriched with negative loadings in adipose tissue (Supplementary Figs. 5, 7 and 8). Expression of genes in both pathways (for which there is some overlap) has been previously associated with insulin sensitivity[12,47,48], suggesting that heritable variation in regulation of these pathways may influence risk of insulin resistance.

Looking at the 10 most positively and negatively loaded transcripts from each tissue, it is apparent that transcripts in the adipose tissue had the largest loadings, both positive and negative (Fig. 5A bar plot). This suggests that much of the effect of genetics on body weight and insulin reisistance is mediated through gene expression in adipose tissue. This finding does not speak to the relative importance of tissues not included in this study, such as brain, in which transcriptional variation may mediate a large portion of the genetic effect on obesity[49]. The strongest loadings in liver and pancreas were comparable, and those in skeletal muscle were the weakest (Fig. 5A), suggesting that less of the genetic effects were mediated through transcription in skeletal muscle. Heritability analysis showed that transcripts with the largest loadings had higher distal heritability (31.8%) than local heritability (5%) (Fig. 5A) (two-sided Welch's t-test $t = 16.4$; df = 100.2; difference 95% CI = 0.24 to 0.30; $p < 2. 2^{-16}$). We also performed TWAS in this population by imputing transcript levels for each gene based on local genotype only and correlating the imputed transcript levels with each trait. In contrast to HDMA, the TWAS procedure (Fig. 5B) tended to nominate transcripts with lower loadings, higher local heritability (15%), and lower distal heritability (20%) (two-sided Welch's t-test $t = 1.9$; df = 151.7; difference 95% CI = −0.002 to 0.1; $p = 0.77$). Finally, we focused on transcripts with the highest local heritability in each tissue (Fig. 5C). This procedure selected transcripts with low loadings on average, consistent with our findings above that high local heritability was associated with low trait correlations (Fig. 2B).

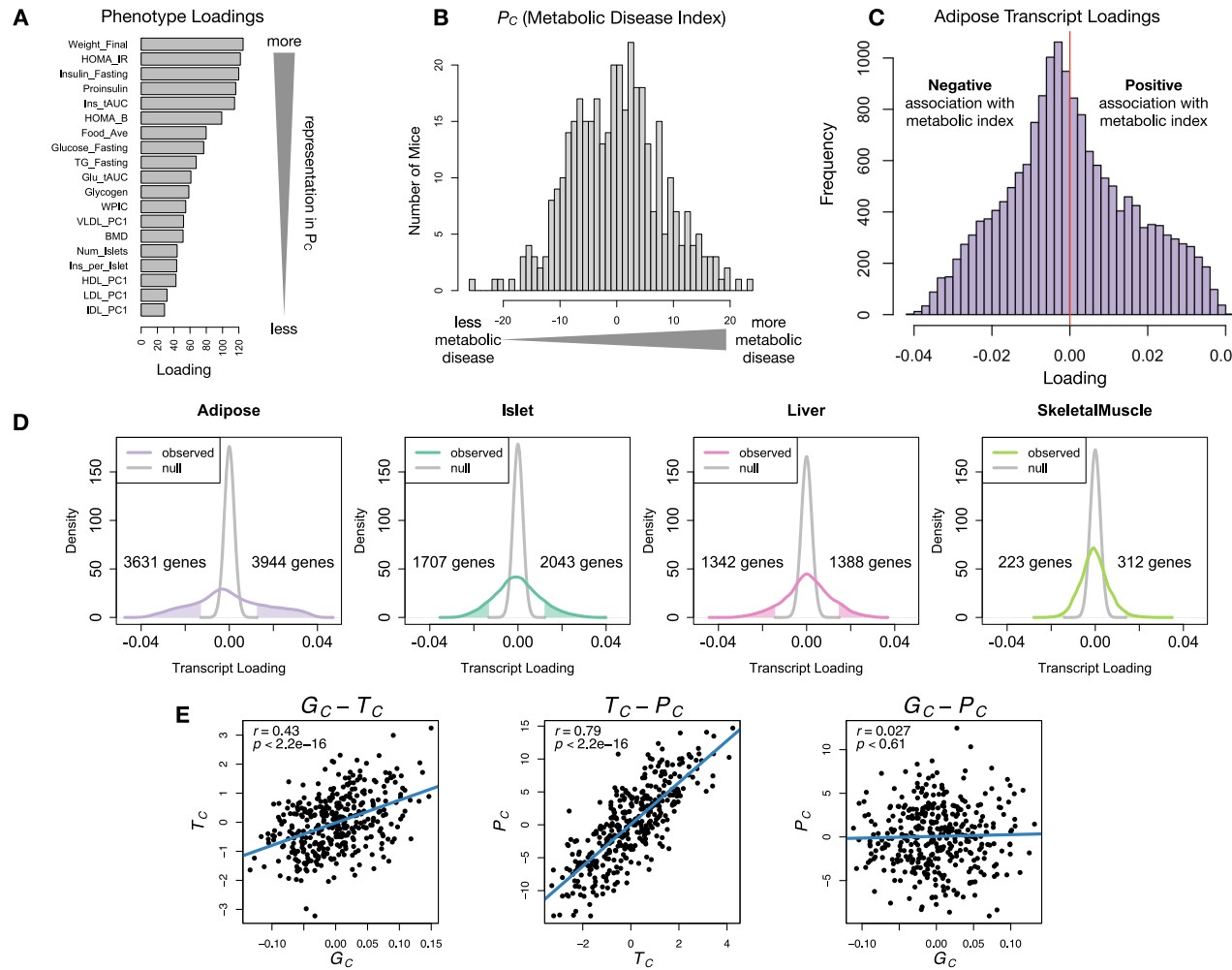

**Fig. 4 | Interpretation of loadings. A.** Loadings across traits. Body weight and insulin resistance contributed the most to the composite trait. **B.** Phenotype scores across individuals. Individuals with large positive phenotype scores had higher body weight and insulin resistance than average. Individuals with large negative phenotype scores had lower body weight and insulin resistance than average. **C.** Distribution of transcript loadings in adipose tissue (purple). For transcripts with large positive loadings, higher expression was associated with higher phenotype scores. For transcripts with large negative loadings, higher expression was associated with lower phenotype scores. **D.** Distributions of loadings across tissues compared to null distributions. Shaded areas represent loadings that were more extreme than the null distribution. Numbers indicate how many transcripts had loadings above and below the extremes of the null. Transcripts in adipose tissue

(purple) had the most extreme loadings indicating that transcripts in adipose tissue were the best mediators of the genetic effects on body weight and insulin resistance. **E.** Scatter plots showing correlations between composite vectors for the genome ($G_C$), the transcriptome ($T_C$), and the phenome ($P_C$). The $G_C$ - $T_C$ association was significant (Linear regression $R^2 = 0.18$; beta coefficient = $7.8 \pm 0.86$ standard error; $t = 9.03$; $p < 2. 2^{-16}$). The $T_C$ - $P_C$ association was significant (Linear regression $R^2 = 0.62$; beta coefficient = $3.1 \pm 0.13$ standard error; $t = 24.4$; $p < 2. 2^{-16}$). There is no association between $G_C$ and $P_C$ (Linear regression $R^2 = 7.1 \times 10^4$; beta coefficient = $2.0 \pm 3.8$ standard error; $t = 0.51$; $p = 0.61$). This correlation structure is consistent with perfect mediation. Blue lines show lines of best fit. Source data are provided as a Source Data file.

We performed a literature search for the genes in each of these groups along with the terms diabetes, obesity, and the name of the expressing tissue to determine whether any of these genes had previous tissue-specific associations with metabolic disease in the literature (Methods). Multiple genes in each group had been previously associated with obesity and diabetes in the represented tissue (Fig. 5 bolded gene names). Genes with high loadings were most highly enriched for previous literature support. They were 2.2 times more likely than TWAS hits and 4 times more likely than genes with high local heritability to be previously associated with obesity or diabetes.

### Tissue-specific transcriptional programs were associated with metabolic traits

Clustering of transcripts with top loadings in each tissue showed tissue-specific functional modules associated with obesity and insulin resistance (Fig. 6A) (Methods). The clustering highlights the

importance of immune activation, particularly in adipose tissue. The mitosis cluster had large positive loadings in three of the four tissues potentially suggesting system-wide proliferation of immune cells. Otherwise, all clusters were strongly loaded in only one or two tissues. For example, the lipid metabolism cluster was loaded most heavily in liver. The positive loadings suggest that high expression of these genes, particularly in the liver, was associated with increased metabolic disease. This cluster included the gene *Pparg*, whose primary role is in the adipose tissue where it is considered a master regulator of adipogenesis[50]. Agonists of *Pparg*, such as thiazolidinediones, are FDA-approved to treat type II diabetes, and reduce inflammation and adipose hyptertrophy[50]. Consistent with this role, the loading for *Pparg* in adipose tissue was negative, suggesting that higher expression was associated with leaner mice (Fig. 6B). In contrast, *Pparg* had a large positive loading in liver (Fig. 6B), where it is known to play a role in the development of hepatic steatosis, or fatty liver. Mice that lack *Pparg*

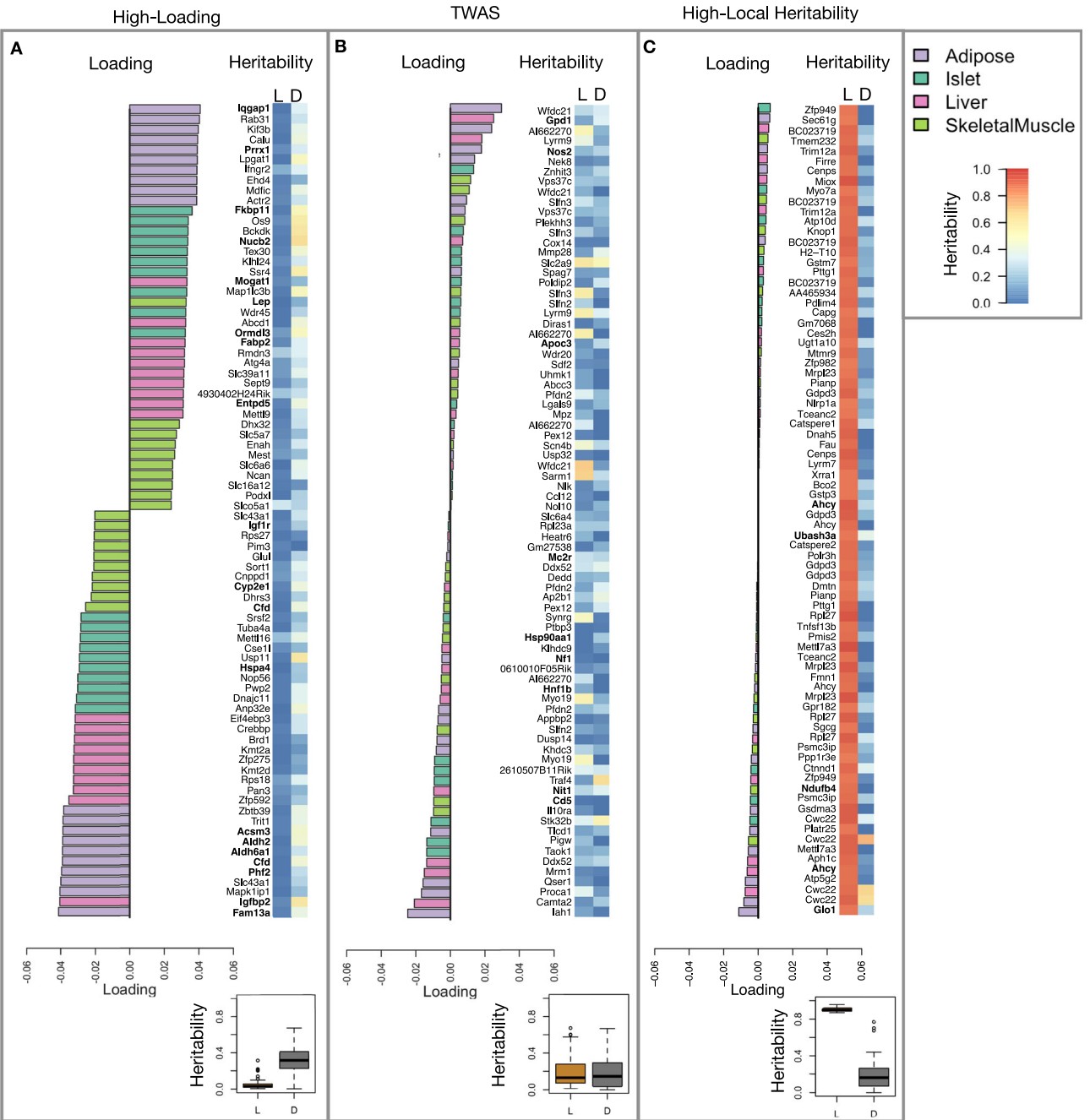

**Fig. 5 | Transcripts with high loadings have high distal heritability and literature support.** Each panel has a bar plot showing the loadings of transcripts selected by different criteria. Bar color indicates the tissue of origin. The heat map shows the local (L - left) and distal (D - right) heritability of each transcript. **A** Loadings for the 10 transcripts with the largest positive loadings and the 10 transcripts with the largest negative loadings for each tissue. Mean distal heritability (31.8%) was significantly higher than mean local heritability (5%) (two-sided Welch's t-test $t = 16.4$; df = 100.2; difference 95% CI = 0.24 to 0.30; $p < 2. 2^{-16}$). **B** Loadings of TWAS candidates with the 10 largest positive correlations with traits and the largest negative correlations with traits across all four tissues. Mean local (15%) and distal (20%) heritability were not significantly different for this group of transcripts (two-sided Welch's t-test $t = 1.9$; df = 151.7; difference 95% CI = −0.002 to 0.1; $p = 0.77$). **C** The transcripts with the largest local heritability (top 20) across all four tissues. Mean local heritability (90%) was significantly higher than mean distal heritability (15%) of these genes (two-sided Welch's $t = 45.0$; df = 82.0; difference 95% CI = 0.72 to 0.78; $p < 2. 2^{-16}$). Lines in boxes correspond to the median; lower and upper edges of boxes indicate the first and third quartiles; whiskers indicate the first and third quartiles ± 1.5 times the interquartile range; dots indicate outliers beyond 1.5 times the interquartile range. All $p$ values derived from two-sided Welch's t-test and are not adjusted for multiple comparisons. Source data are provided as a Source Data file.

specifically in the liver, are protected from developing steatosis and show reduced expression of lipogenic genes[51,52]. Overexpression of *Pparg* in the livers of mice with a *Ppara* knockout, causes upregulation of genes involved in adipogenesis[53]. In the livers of both mice and humans high *Pparg* expression is associated with hepatocytes that accumulate large lipid droplets and have gene expression profiles

similar to that of adipocytes[54,55]. The local and distal heritability of *Pparg* is low in adipose tissue suggesting its expression in this tissue is highly constrained in the population (Fig. 6B). However, the distal heritability of *Pparg* in liver is relatively high suggesting it is complexly regulated and has sufficient variation in this population to drive variation in phenotype. Both local and distal heribatility of *Pparg* in the

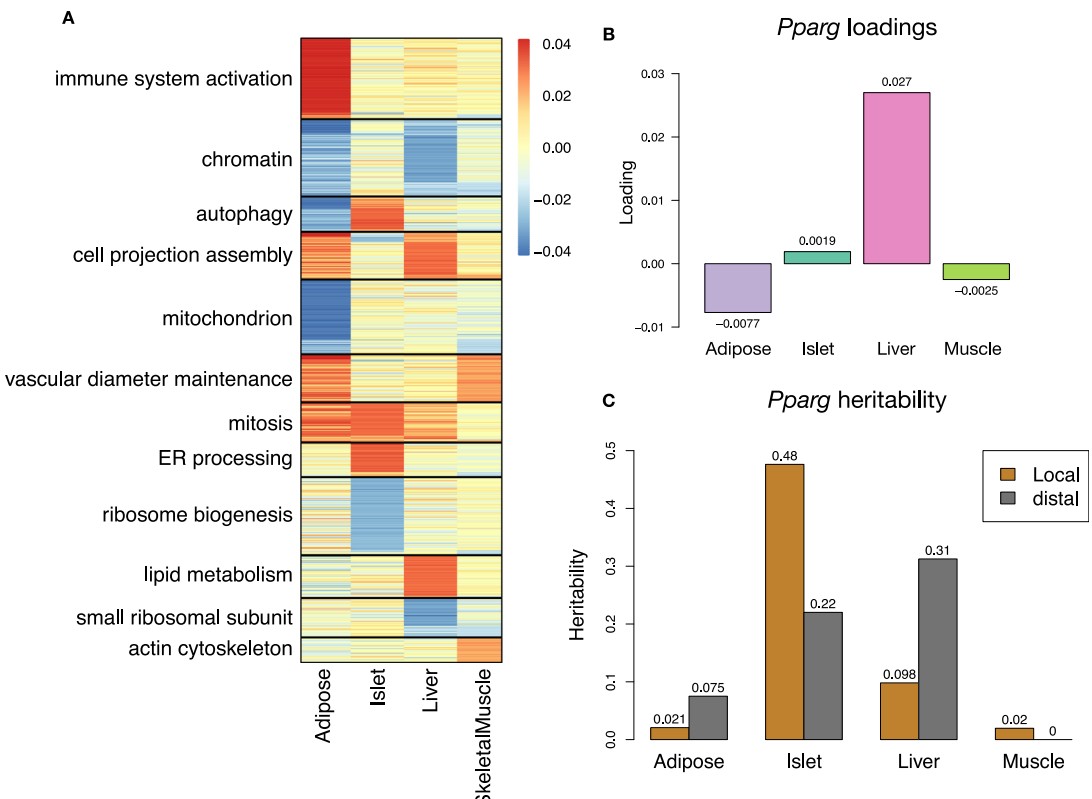

**Fig. 6 | Tissue-specific transcriptional programs are associated with obesity and insulin resistance. A** Heat map showing the loadings of all transcripts with loadings greater than 2.5 standard deviations from the mean in any tissue. The heat map was clustered using *k* medoid clustering. Functional enrichments of each cluster are indicated along the left margin. **B** Loadings for *Pparg* in different tissues indicated by color. **C** Local (brown) and distal (gray) of *Pparg* expression in different tissues. Source data are provided as a Source Data file.

islet are relatively high, but the loading is low, suggesting that variability of expression in the islet does not drive variation in MDI. These results highlight the importance of tissue context when investigating the role of heritable transcript variability in driving phenotype. Gene lists for all clusters with tissue-specific loadings are available in Supplementary Data 1.

### Gene expression, but not local eQTLs, predicted body weight in an independent population

To test whether the transcript loadings identified in the DO could be translated to another population, we tested whether they could predict metabolic phenotypes in an independent population of CC-RIX mice, which were F1 mice derived from multiple pairings of Collaborative Cross (CC)[32,56–58] strains (Fig. 7) (Methods). We asked whether the loadings identified in the DO mice were relevant to the relationship between the transcriptome and the phenome in the CC-RIX. We predicted body weight (a surrogate for MDI) in each CC-RIX individual using measured gene expression in each tissue and the transcript loadings identified in the DO (Methods). The predicted body weight and acutal body weight were highly correlated (Fig. 7B). The best prediction was achieved for adipose tissue, which supports the observation in the DO that adipose expression was the strongest mediator of the genetic effect on MDI. This result also confirms the validity and translatability of the transcript loadings and their relationship to metabolic disease.

We then investigated the source of the relevant variation in gene expression. If local regulation was the predominant factor influencing trait-relevant gene expression, we should be able to predict phenotype in the CC-RIX using transcripts imputed from local genotype (Fig. 7A). The DO and the CC-RIX were derived from the same eight founder strains and so carry the same alleles throughout the genome. We imputed gene expression in the CC-RIX using local genotype and were able to estimate variation in gene transcription robustly (Supplementary Fig. 9). However, these imputed values failed to predict body weight in the CC-RIX when weighted with the loadings from HDMA. (Fig. 7C). This result suggests that local regulation of gene expression is not the primary factor driving heritability of complex traits. It is also consistent with our findings in the DO population that distal heritability was a major driver of trait-relevant gene expression and that high-loading transcripts had comparatively high distal, and low local, heritability.

### Distally heritable transcriptomic signatures suggested variation in composition of adipose tissue and islets

The interpretation of global genetic influences on gene expression and phenotype is potentially more challenging than the interpretation and translation of local genetic influences, as genetic effects cannot be localized to individual gene variants or transcripts. However, there are global patterns across the loadings that can inform mechanism. For example, heritable variation in cell type composition can be inferred from transcript loadings. We observed above that immune activation in the adipose tissue was a highly enriched process correlating with obesity in the DO population. In humans, it has been extensively observed that macrophage infiltration in adipose tissue is a marker of obesity and metabolic disease[59]. To determine whether the immune activation reflected a heritable change in cell composition in adipose tissue in DO mice, we compared loadings of cell-type specific genes in adipose tissue (Methods). The mean loading of macrophage-specific genes was significantly greater than 0 (Holm-adjusted two-sided empirical $p < 2 \times 10^{-16}$) (Fig. 8A), indicating that obese mice were

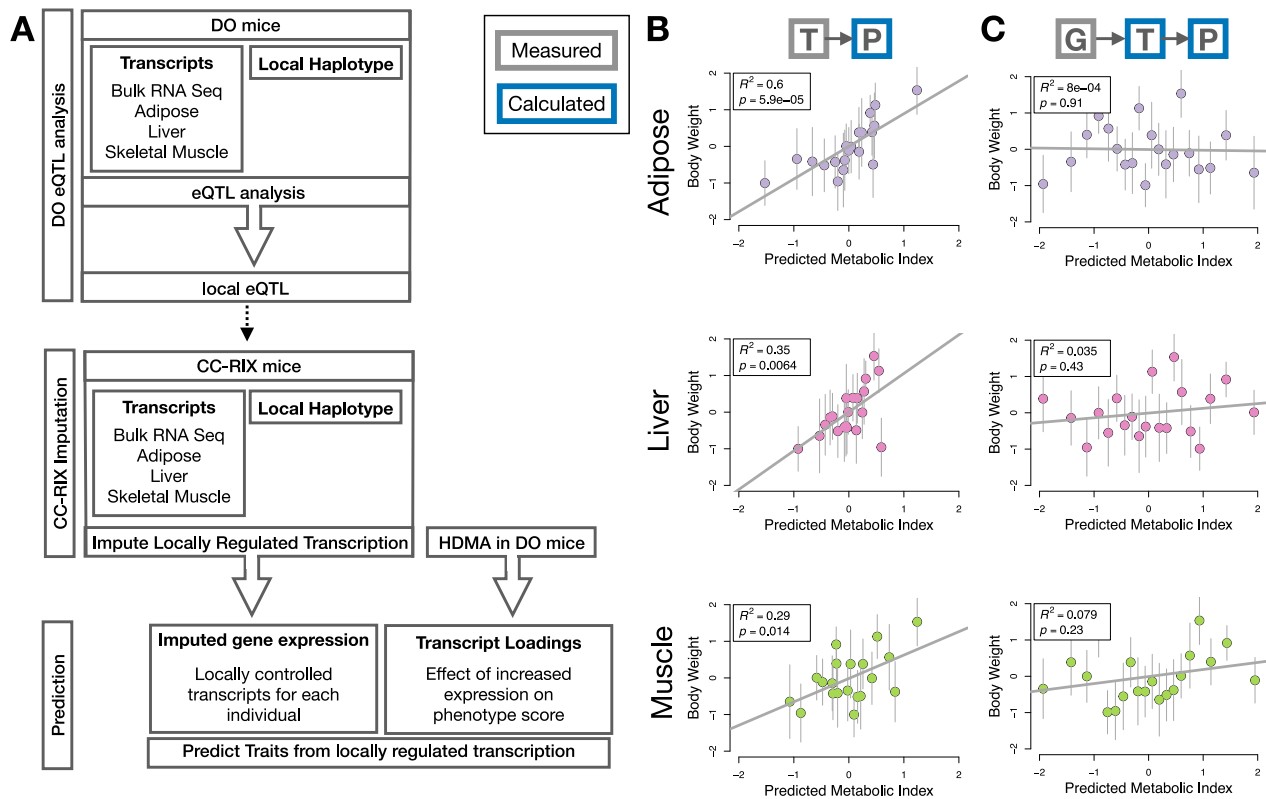

**Fig. 7 | Transcription, but not local genotype, predicts phenotype in the CC-RIX. A** Workflow showing procedure for translating HDMA results to an independent population of mice. **B** Relationships between the predicted metabolic disease index (MDI) and the mean of the rank normal body weight. In this column, MDI was derived from measured transcripts. Adipose: $R^2 = 0.60$; beta coefficient = $0.89 \pm 0.17$ standard error; $t = 5.21$; $p = 5.9 \times 10^{-5}$. Liver: $R^2 = 0.35$; beta coefficient = $1.1 \pm 0.34$ standard error; $t = 3.1$; $p = 6.4 \times 10^{-3}$. Muscle: $R^2 = 0.29$; beta coefficient = $0.64 \pm 0.24$ standard error; $t = 2.7$; $p = 0.014$. **C** In this column, MDI was derived from transcripts imputed from local genotype. Adipose: $R^2 = 8.0 \times 10^{-4}$;

beta coefficient = $-0.2 \pm 0.16$ standard error; $t = -0.12$; $p = 0.91$. Liver: $R^2 = 0.035$; beta coefficient = $0.13 \pm 0.16$ standard error; $t = 0.81$; $p = 0.43$. Muscle: $R^2 = 0.079$; beta coefficient = $0.19 \pm 0.16$ standard error; $t = 1.24$; $p = 0.23$. Gray boxes indicate measured quantities and blue boxes indicate calculated quantities. G - genome; T - transcriptome; P - phenome (here MDI). The dots in each panel represent individual CC-RIX strains. Each strain was represented by between 19 and 24 individuals. The gray lines show the standard deviation of mean body weight for the strain. Source data are provided as a Source Data file.

genetically predisposed to have high levels of macrophage infiltration in adipose tissue in response to the HFHS diet. Loadings for marker genes for other cell types were not statistically different from zero (Adipocytes: $p = 0.08$, Progenitors: $p = 0.58$, Leukocytes: $p = 0.28$; all Holm-adjusted two-sided empirical $p$), indicating that changes in the abundance of those cell types was not a mediator of MDI.

We also compared loadings of cell-type specific transcripts in islet (Methods). The mean loadings for alpha-cell specific transcripts were significantly greater than 0 (Holm-adjusted two-sided empirical $p = 0.002$), while the mean loadings for transcripts specific to delta cells (Holm-adjusted two-sided empirial $p < 2 \times 10^{-16}$) and endothelial cells (Holm-adjusted two-sided empirical $p = 0.01$) were significantly less than 0 (Fig. 8B). These results suggest that mice with higher MDI inherited an altered cell composition that predisposed them to metabolic disease, or that these compositional changes were induced by the HFHS diet in a heritable way. In either case, these results support the hypothesis that alterations in islet composition drive variation in MDI. Notably, the mean loading for pancreatic beta cell marker transcripts was not significantly different from zero (Holm-adjusted two-sided empirical $p = 0.95$). We stress that this is not necessarily reflective of the function of the beta cells in the obese mice, but rather suggests that any variation in the number of beta cells in these mice was unrelated to obesity and insulin resistance, the major contributors to MDI. This is further consistent with the islet composition traits having small loadings in the phenome score (Fig. 4).

### Heritable transcriptomic signatures translated to human disease

Ultimately, the heritable transcriptomic signatures that we identified in DO mice will be useful if they inform mechanism and treatment of human disease. To investigate the potential for translation of the gene signatures identified in DO mice, we compared them to transcriptional profiles in obese and non-obese human subjects (Methods). We limited our analysis to adipose tissue because the adipose tissue signature had the strongest relationship to obesity and insulin resistance in the DO.

We calculated a predicted MDI for each individual in the human studies based on their adipose tissue gene expression (Methods) and compared the predicted scores for obese and non-obese groups as well as diabetic and non-diabetic groups. In all cases, the predicted MDIs were higher on average for individuals in the obese and diabetic groups compared with the lean and non-diabetic groups (Fig. 8D). This indicates that the distally heritable signature of MDI identified in DO mice is relevant to obesity and diabetes in human subjects.

### Existing therapies are predicted to target mediator gene signatures

Another application of the transcript loading landscape is in ranking potential drug candidates for the treatment of metabolic disease. Although high-loading transcripts may be good candidates for understanding specific biology related to obesity, the transcriptome overall is highly interconnected and redundant. The ConnectivityMap (CMAP)

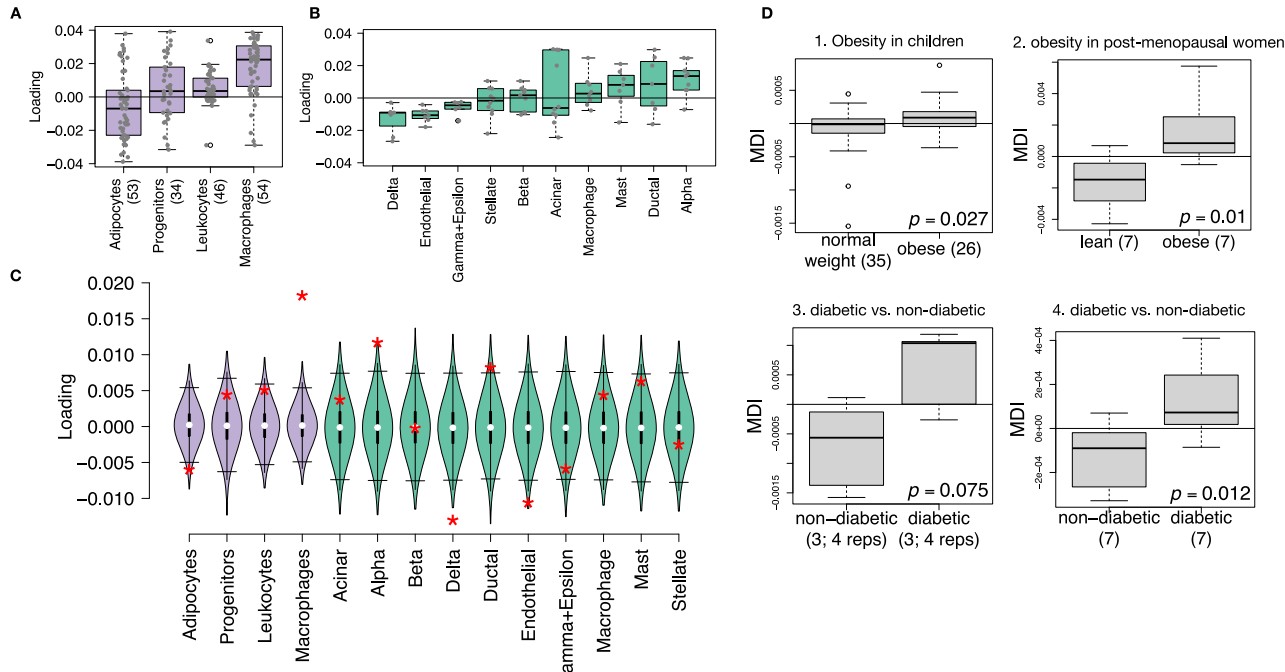

**Fig. 8 | HDMA results translate to humans. A** Distribution of loadings for cell-type-specific transcripts in adipose tissue (purple). Numbers in parentheses indicate the number of transcripts in each group. **B** Distribution of loadings for cell-type-specific transcripts in pancreatic islets (green). Each box in this panel represents 10 transcripts. **C** Null distributions from 10,000 permutations for the mean loading of randomly selected transcripts in each cell type compared with the observed mean loading of each group of transcripts (red asterisk). Violin plot colors indicate the tissue of each cell type and match (**A** and **B**) (purple = adipose; green = islet) (**D**). Predictions of metabolic disease index (MDI) in four adipose transcription data sets downloaded from GEO. In each study the obese/diabetic patients were predicted to have greater MDI than the lean/non-diabetic patients based on the HDMA results from DO mice. 1) two-sided Welch's $t$ test: $t = 2.28$, df = 58.1, 95%

CI of difference = $2.1 \times 10^{-5}$ to $3.2 \times 10^{-4}$ A.U.; $p = 0.027$ 2) two-sided Welch's $t$ test: $t = 3.04$, df = 11.84, 95% CI of difference = $9.3 \times 10^{-4}$ to $5.7 \times 10^{-3}$ A.U.; $p = 0.01$ 3) linear mixed effects model: fixed effect diabetic = $1.4 \times 10^{-3} \pm 5.6 \times 10^{-4}$ Std. Error; $t = 2.4$, df = 4, $p = 0.075$ 4) two-sided Welch's $t$ test: $t = 2.95$, df = 11.89, 95% CI of difference = $6.8 \times 10^{-5}$ to $4.6 \times 10^{-4}$ A.U.; $p = 0.012$ Lines in boxes correspond to the median; lower and upper edges of boxes indicate the first and third quartiles; whiskers indicate either the minimum and maximum values if no outliers, or the first and third quartiles $\pm 1.5$ times the interquartile range if there are outliers; dots indicate outliers beyond 1.5 times the interquartile range. The number of patients in each group is indicated by numbers in parentheses with the number of technical replicates (reps) if relevant. No adjustments were made for multiple comparisons. Source data are provided as a Source Data file.

database[60,61] developed by the Broad Institute allows querying thousands of compounds that reverse or enhance the extreme ends of transcriptomic signatures in multiple different cell types. By identifying drugs that reverse pathogenic transcriptomic signatures, we can potentially identify compounds that have favorable effects on gene expression. To test this hypothesis, we queried the CMAP database through the CLUE online query tool (https://clue.io/query/, version 1.1.1.43) (Methods). We identified top anti-correlated hits across all cell types (Supplementary Figs. 10 and 11). To get more tissue-specific results, we also looked at top results in cell types that most closely resembled our tissues. We looked at results in adipocytes (ASC) as well as pancreatic tumor cells (YAPC) regardless of $p$ value (Supplementary Figs. 12 and 13).

The CMAP database identified both known diabetes drugs (e.g., sulfonylureas), as well as drugs that target pathways known to be involved in diabetes pathogenesis (e.g., mTOR inhibitors). These findings help support the mediation model we fit here. Although the composite variables we identified here are consistent with mediation, they do not prove causality. However, the results from CMAP suggest that reversing the transcriptomic signatures we found also reverses metabolic disease phenotypes, which supports a causal role of the transcript levels in driving pathogenesis of metabolic disease. These results thus support the mediation model we identified here and its translation to therapies in human disease.

## Discussion
Here we investigated the relative contributions of local and distal gene regulation in four tissues to heritable variation in traits related to

metabolic disease in genetically diverse mice. We found that distal heritability was positively correlated with trait relatedness, whereas high local heritability was negatively correlated with trait relatedness. We used a novel high-dimensional mediation analysis (HDMA) to identify tissue-specific composite transcripts that are predicted to mediate the effect of genetic background on metabolic traits. The adipose-derived composite transcript robustly predicted body weight in an independent cohort of diverse mice with a disparate population structure. It also predicted MDI in four human cohorts. However, gene expression imputed from local haplotype failed to predict body weight in the second mouse population. Taken together, these results highlight the complexity of gene expression regulation in relation to trait heritability and suggest that heritable trait variation is mediated primarily through distal gene regulation.

Our result that distal regulation accounted for most trait-related gene expression differences is consistent with a complex model of genetic trait determination. It has frequently been assumed that gene regulation in *cis* is the primary driver of genetically associated trait variation, but attempts to use local gene regulation to explain phenotypic variation have had limited success[16,17]. In recent years, evidence has mounted that distal gene regulation may be an important mediator of trait heritability[18,19,62,63]. It has been observed that transcripts with high local heritability explain less expression-mediated disease heritability than those with low local heritability[19]. Consistent with this observation, genes located near GWAS hits tend to be complexly regulated[18]. They also tend to be enriched with functional annotations, in contrast to genes with simple local regulation, which tend to be depleted of

functional annotations suggesting they are less likely to be directly involved in disease traits[18]. These observations are consistent with principles of robustness in complex systems in which simple regulation of important elements leads to fragility of the system[64–66]. Our results are consistent, instead, with a more complex picture where genes whose expression can drive trait variation are buffered from local genetic variation but are extensively influenced indirectly by genetic variation in the regulatory networks converging on those genes.

Our results are also consistent with the recently proposed omnigenic model, which posits that complex traits are massively polygenic and that their heritability is spread out across the genome[67]. In the omnigenic model, genes are classified either as core genes, which directly impinge on the trait, or peripheral genes, which are not directly trait-related, but influence core genes through the complex gene regulatory network. HDMA explicitly models a central proposal of the omnigenic model which posits that once the expression of the core genes (i.e., trait-mediating genes) is accounted for, there should be no residual correlation between the genome and the phenome. Here, we were able to fit this model and identified a composite transcript that, when taken into account, left no residual correlation between the composite genome and composite phenome scores (Figs. 3A, and 4E).

Unlike in the omnigenic model, we did not observe a clear demarcation between the core and peripheral genes in loading magnitude, but we do not necessarily expect a clear separation given the complexity of gene regulation and the genotype-phenotype map[68].

An extension of the omnigenic model proposed that most heritability of complex traits is driven by weak distal eQTLs that are potentially below the detection threshold in studies with feasible sample sizes[62]. This is consistent with what we observed here. For example, the gene *Nucb2*, had a high loading in islets and was also strongly distally regulated (65% distal heritability) (Fig. 5). This gene is expressed in pancreatic $\beta$ cells and is involved in insulin and glucagon release[69–71]. Although its transcription was highly heritable in islets, that regulation was distributed across the genome, with no clear distal eQTL (Supplementary Fig. 14). Thus, although distal regulation of some genes may be strong, this regulation is likely to be highly complex and not easily localized.

Individual high-loading transcripts also demonstrated biologically interpretable, tissue-specific patterns. We highlighted *Pparg*, which is known to be protective in adipose tissue[50] where it was negatively loaded, and harmful in the liver[51–55], where it was positively loaded. Such granular patterns may be useful in generating hypotheses for further testing, and prioritizing genes as therapeutic targets. The tissue-specific nature of the loadings also may provide clues to tissue-specific effects, or side effects, of targeting particular genes system-wide.

In addition to identifying individual transcripts of interest, the composite transcripts can be used as weighted vectors in multiple types of analysis, such as drug prioritization using gene set enrichment analysis (GSEA) and the CMAP database. In particular, the CMAP analysis identified drugs that have been demonstrated to reverse insulin resistance and other aspects of metabolic disease. This finding supports the hypothesis that HDMA identified transcripts that truly mediate genetic effects on traits. HDMA identifies transcriptional patterns that are consistent with a mediation model, but alone does not prove mediation. However, the finding that these drugs act both on the transcriptional patterns and on the desired traits support the mediation model and the hypothesis that these transcripts play a causal role in pathogenesis of metabolic disease.

Together, our results have shown that both tissue specificity and distal gene regulation are critically important to understanding the genetic architecture of complex traits. We identified important genes and gene signatures that were heritable, plausibly causal of disease, and translatable to other mouse populations and to humans. Finally, we have shown that by directly acknowledging the complexity of both gene regulation and the genotype-to-phenotype map, we can gain a

new perspective on disease pathogenesis and develop actionable hypotheses about pathogenic mechanisms and potential treatments.

## Methods

### Statistics and reproducibility

In this study we used two populations of laboratory mice (*Mus musculus*): 1) a population of 482 diversity outbred mice (J:DO, JAX strain#:009376) (*n* = 240 females and *n* = 242 males), and 2) a population of 466 CC-RIX mice (*n* = 234 females, and *n* = 232 males), which were F1 mice derived from crosses of the following JAX strains of Collaborative Cross mice: CC043/GeniUncJ (strain#:023828), CC011/UncJ (strain#018854), CC030/GeniUncJ (strain#:025426), CC002/UncJ (strain#:021236), CC051/TauUncJ (strain#:021897), CC019/TauUncJ (strain#:CC019), CC027/GeniUncJ (strain#:025130), CC012/GeniUncJ (strain#:028409), CC024/GeniUncJ (strain#:021891), CC075/UncJ (strain#:027293), CC005/TauUncJ (strain#:020945), CC059/TauUncJ (strain#:025125), CC001/UncJ (strain#:021238), CC042/GeniUncJ (strain#:020947), CC040/TauUncJ (strain#:023831), CC004/TauUncJ (strain#:020944), CC009/UncJ (strain#:018856), CC013/GeniUncJ (strain#:021892), and CC060/UncJ (strain#:026427). Counts of male and female mice in each strain are available in the Supplementary Data 2. All DO mice were maintained on a high-fat, high-sugar diet. The CC-RIX mice were randomized to a high-fat or a low-fat diet. The diets were different colors, so blinding was not possible. CC-RIX mice were sacrificed in a 6-month cohort and a 12-month cohort. In the DO population, we used sex, DO generation, and DO wave as covariates in all statistical tests. In the CC-RIX population, we used diet, sex, and age as covariates in all statistical tests. We did not investiagte sex-specific effects on genetic architecture in order to maximize power and generalizability, as well as to simplify the overall analysis. CC-RIX animals were randomly assigned to housing and diets based on litters with multiple litters in each group. While undergoing metabolic phenotyping animals were randomly assigned an order for testing. Sample sizes were determined using statistical rules of thumb[72]. We included at least four biological replicates that were tested over multiple months to increase replication power. No data were excluded. Further experimental details and statistical testing are described individually in the following Methods sections.

### Diversity outbred mice

Mice were maintained and treated in accordance with the guidelines approved by the Department of Biochemistry animal vivarium at the University of Wisconsin. Animal husbandry and in vivo phenotyping methods were previously published are are described briefly below[12,28].

A population of 482 diversity outbred mice (split evenly between male and female) from generations 18, 19, and 21, was placed on a high-fat (44.6% kcal fat), high-sugar (34% carbohydrate), adequate protein (17.3% protein) diet from Envigo Teklad (catalog number TD.08811) starting at four weeks of age as described previously[12]. Individuals were assessed longitudinally for multiple metabolic measures including fasting glucose levels, glucose tolerance, insulin levels, body weight, and blood lipid levels.

When mice were harvested at 22 weeks of age, their pancreatic islets were isolated by hand. Insulin per islet was measured, and whole pancreas insulin content was calculated from the insulin per islet measure and the total numer of islets per pancreas[12]. RNA was isolated from the whole islets and sent to The Jackson Laboratory for high-throughput sequencing[12].

**Trait measurements.** Trait measurements were described previously in Keller et al. 2018[12]. Briefly, body weight was measured every two weeks, and 4-h fasting plasma samples were collected to measure insulin, glucose, and triglycerides (TG). At around 18 weeks of age, an oral glucose tolerance test (oGTT) was conducted on 4-hour fasted mice to assess changes in plasma insulin and glucose. Glucose (2 g/kg)

was given via oral gavage. Blood samples were taken from a retro-orbital bleed before glucose administration, and at 5, 15, 30, 60, and 120 minutes afterward. The area under the curve (AUC) was calculated for glucose and insulin. Glucose was measured using the glucose oxidase method, and insulin was measured by radioimmunoassay.

HOMA-IR and HOMA-B, which are homeostatic model assessments of insulin resistance (IR) and pancreatic islet function (B), were calculated using fasting plasma glucose and insulin values at the start of the oGTT. HOMA-IR = (glucose × insulin) / 405 and HOMA-B = (360 × insulin) / (glucose - 63). Plasma glucose and insulin units are mg/dL and mU/L, respectively.

**Genotyping.** Genotypes at 143,259 markers was performed using the Mouse Universal Genotyping Array (GigaMUGA)[73] at Neogen (Lincoln, NE) as described previously[12,74]. Genotypes were converted to founder strain-haplotype reconstructions using the R/DOQTL software[75] and interpolated onto a grid with 0.02-cM spacing to yield 69,005 pseudo-markers. Individual chromosome (Chr) haplotypes were reconstructed from RNA-seq data using a hidden Markov model[76] (GBRS, https://github.com/churchill-lab/gbrs). Using both methods to call haplotypes provided redundancy for quality control. Three mice had inconsistent calls between the two methods and were excluded from the analysis[12].

**Trait selection in DO.** We filtered the measured traits in this study to a set of relatively non-redundant measures that were well-represented in the population (having at least 80% of individuals measured). A complete description of trait filtering can be found at Figshare https://doi.org/10.6084/m9.figshare.27066979[77] in the file Documents > 1.DO > 1b.Trait_Selection.Rmd.

We took two approaches for traits with multiple redundant measurements, for example longitudinal body weights. In the case of longitudinal measurements, we used the final measurement, as this was the closest physiological measurement to the measurement of gene expression, which was done at the end of the experiment. The labels for these traits have the word Final appended to their name. For traits with multiple highly related measurements, such as cholesterol, we used the first principal component of the group of measurements. For example, we used the first principal component of all LDL measurements as the measurement of LDL. For each set of traits, we ensured the first principal component had the correct sign by correlating it with the average of the traits. For pearson correlation coefficients (*r*) less than 0, we multiplied the principal component by −1. The labels for these traits have the term PC1 appended to their name.

Heritability of each trait was estimates using the R package qtl2[34]. Standard errors of heritability were estimated using a purpose-built function by Karl Broman available on GitHub [https://github.com/rqtl/qtl2/issues/193]. This method is based on Equation 2 in Visscher and Goddard (2015)[78].

**Processed DO data.** The DO data used in this study were generated in a previous study[12,28]. We downloaded genotypes, phenotypes, and pancreatic islet gene expression data from Dryad https://doi.org/10.5061/dryad.pj105[79].

**Collaborative cross recombinant inbred mice (CC-RIX)**
CC-RIX mice were derived by crossing strains of Collaborative Cross (CC) mice to produce heterozygous F1 mice. Each strain included between 19 and 24 biological replicates. The strains used and the number of mice in each are reported in the supplemental file CC-RIX_strain_table.txt.

Mice were cared for and treated following the guidelines approved by the Association for Assessment and Accreditation of Laboratory Animal Care at The Jackson Laboratory. All animals were obtained from The Jackson Laboratory. The mice were kept in a pathogen-free room at a temperature ranging from 20 to 22 °C with a 12-h light/dark cycle.

Starting at 6 weeks of age, they were fed either a custom-designed high-fat, high-sugar (HFHS) diet (Research Diets D19070208) or a control diet (Research Diets D19072203) *ad libitum*. Body weight was measured weekly until the mice were about 16 weeks old, after which measurements were taken every other week. Food intake measurements were collected at 14 weeks, 23 weeks (for 6-month cohorts), 26 weeks (for 12-month cohorts), 38 weeks, and 51 weeks by weighing the grain contents in the cage over a three-day period. Fasted serum was collected at 14 weeks, 28 weeks (for 6-month cohorts), 26 weeks (for 12-month cohorts), 38 weeks, and 56 weeks of age via retro-orbital or submental vein. Sex, diet, and age were used as covariates in all analyses.

**CC-RIX genotypes**
We used the most recent common ancestor (MRCA) genotypes for the Collaborative Cross (CC) mice available on the University of North Carolina Computational Systems Biology website: http://www.csbio.unc.edu/CCstatus/CCGenomes/.

To generate CC-RIX genotypes, we averaged the haplotype probabilities for the two parental strains at each locus.

**Clinical chemistries.** CC-RIX animals were fasted for four hours before serum collection via the retro-orbital or submental vein. Whole blood was left at room temperature for 30–60 min before being centrifuged for 5 minutes at 14,674 g. The serum was then tested for glucose (Beckman Coulter; OSR6121), cholesterol (Beckman Coulter; OSR6116), triglycerides (Beckman Coulter; OSR60118), insulin (MSD; K152BZC-1), or c-peptide (MSD; K1526JK-1).

**Intraperitoneal glucose tolerance testing.** After a fasting period of 4–6 h, baseline glucose measurements were taken from CC-RIX mice using an AlphaTrak2 glucometer and test strips (Zoetis) by making a small nick in the tail tip. A bolus intraperitoneal injection of 20% glucose (1g/kg) was then administered, and additional tail tip nicks were performed at 15, 30, 60, and 120 minutes post-injection to measure glucose levels.

**Bulk tissue collection.** At either 28 weeks of age (for the 6-month cohort) or 56 weeks of age (for the 12-month cohort), CC-RIX animals were humanely euthanized by cervical dislocation. Tissues, including visceral adipose (gonadal fat pad), skeletal muscle (gastrocnemius), and the left liver lobe, were harvested and flash-frozen in liquid nitrogen for RNA sequencing.

**Whole pancreas insulin content**
The animals were humanely euthanized at 16 weeks of age and the entire pancreas was removed, ensuring no excess fat or mesentery tissue was included. The pancreas tissue was placed in a pre-weighed 20 mL glass scintillation vial containing acid ethanol (75% HPLC grade ethanol (ThermoFisher; A995-4), 1.5% concentrated hydrochloric acid (ThermoFisher; A144-212) in distilled water). The weight of the pancreas was measured for normalization. Using curved scissors, the pancreas was chopped for four minutes, and the samples were stored at −20 °C until all animals were harvested. For insulin measurements, the contents of the scintillation vials were rinsed with 4 mL PBS (Roche; 1666789) with 1% BSA (Sigma; A7888), neutralized with 65 μL 10N NaOH (Fisher; SS255-1), and vortexed for 30 seconds. The samples were then centrifuged at 4 °C for 5 minutes at 452 g. The samples were diluted 5000X in PBS with 1% BSA, and insulin was measured (MSD; K152BZC-1).

**RNA isolation and QC**
RNA from both DO and CC-RIX adipose, gastrocnemius, and left liver lobe tissues was isolated using the MagMAX mirVana Total RNA Isolation Kit (ThermoFisher; A27828) and the KingFisher Flex purification system (ThermoFisher; 5400610). The frozen tissues were pulverized with a Bessman Tissue Pulverizer (Spectrum Chemical) and

homogenized in TRIzol™ Reagent (ThermoFisher; 15596026) using a gentleMACS dissociator (Miltenyi Biotec Inc). After adding chloroform to the TRIzol homogenate, the RNA-containing aqueous layer was extracted for RNA isolation, following the manufacturer's protocol, starting with the RNA bead binding step using the RNeasy Mini kit (Qiagen; 74104). RNA concentrations and quality were assessed using the Nanodrop 8000 spectrophotometer (Thermo Scientific) and the RNA 6000 Pico or RNA ScreenTape assay (Agilent Technologies).

## Library construction

Before library construction, 2 $\mu$L of diluted (1:1000) ERCC Spike-in Control Mix 1 (ThermoFisher; 4456740) was added to 100 ng of each RNA sample. Libraries were then constructed using the KAPA mRNA HyperPrep Kit (Roche Sequencing Store; KK8580) following the manufacturer's protocol. The process involves isolating polyA-containing mRNA using oligo-dT magnetic beads, fragmenting the RNA, synthesizing the first and second strands of cDNA, ligating Illumina-specific adapters with unique barcode sequences for each library, and performing PCR amplification. The quality and concentration of the libraries were evaluated using the D5000 ScreenTape (Agilent Technologies) and the Qubit dsDNA HS Assay (ThermoFisher; Q32851), respectively, according to the manufacturers' instructions.

## Sequencing

Libraries were sequenced on an Illumina NovaSeq 6000 using the S4 Reagent Kit (Illumina; 20028312). All tissues underwent 100 bp paired-end sequencing, aiming for a target read depth of 30 million read pairs.

## Processing of RNA sequencing data

We used the Expectation-Maximization algorithm for Allele Specific Expression (EMASE)[80,81] to quantify multi-parent allele-specific and total expression from RNA-seq data for each tissue. EMASE was performed by the Genotype by RNA-seq (GBRS) software package (https://gbrs.readthedocs.io/en/latest/). In the process, R1 and R2 FASTQ files were combined and aligned to a hybridized (8-way) transcriptome generated for the 8 DO founder strains as single-ended reads. GBRS was also used to reconstruct the mouse genotype probabilities along ~ 69K markers, which was used for confirming genotypes in the quality control (QC) process. For the QC process, we used a Euclidean distances method (developed by Greg Keele - Churchill Lab) to compare the GBRS genotype probabilities between the tissues and the genotype probabilities array for all mice. The counts matrix for each tissue was processed to filter out transcripts with less than one read for at least half of the samples. RNA-seq batch effects were removed by regressing out batch as a random effect and considering sex and generation as fixed effects using lme4 R package[82]. RNA-Seq counts were normalized relative to total read counts using the variance stabilizing transform (VST) as implemented in DESeq2[83] and using rank normal score.

## eQTL analysis

We used the R package qtl2[34] to perform eQTL analysis. We used the rank normal score data and used sex, DO generation, and DO wave as additive covariates. We also used kinship as a random effect. We used permutations to find a LOD threshold of eight for significant QTLs which corresponded to a genome-wide $p$ value of 0.01[84].

To assess whether eQTL were shared across tissues, we considered significant eQTLs within 4Mb of each other to be overlapping. We considered local and distal eQTLs separately. Local eQTL were defined as an eQTL within 4Mb of the transcription start site (TSS) of the encoding gene. Distal eQTL were defined as an eQTL greater than 4Mb from the TSS of the encoding gene.

## Local and distal heritability of transcripts

To estimate local and distal heritability of each transcript, we scaled each normalized transcript to have a variance of 1. We then modeled

this transcript with the local genotype using the fit1() function in the R package qtl2[34]. We used the resulting model to predict the transcript values. The variance of the predicted transcript is its local heritability. We then estimated the heritability of the residual of the model fit. The variance of the residual multiplied by its heritability is the estimate of the distal heritability of the transcript.

We compared local and distal estimates of heritability to measures of trait relevance for each transcript. To calculate trait relevance of a given transcript, we adjusted normalized transcript values for sex, DO wave, and DO generation. We similarly adjusted traits by sex, DO wave, and DO generation. We then calculated all Spearman correlation coefficients ($\rho$) between adjusted traits and adjusted transcripts. The trait relevance of a given tanscript was the maximum absolute correlation coefficient across all traits.

## High-dimensional mediation analysis

In this section we derive the objective function for high-dimensional mediation analysis (HDMA) and present an iterative algorithm to optimize this objective function. Our starting point is the univariate case, where we describe perfect mediation as a constraint on the covariance matrix among variables. We then leverage this constraint to define projections of multivariate data that are maximally consistent with perfect mediation (HDMA). Next, we demonstrate how to kernelize HDMA to limit dimensionality of the model and enable non-linear HDMA models.

**Perfect mediation as a constraint on covariance matrices.** Suppose we have three random variables $x$, $m$, and $y$. Assume they each have unit variance and that they satisfy the following structural equation model (SEM) such that $m$ perfectly mediates the effect of $x$ on $y$:

$$m = \alpha x + \epsilon_m \tag{1}$$

$$y = \beta m + \epsilon_y \tag{2}$$

From these structural equations, we have the model-implied covariance matrix, $\Sigma$, given by

$$\Sigma = \begin{bmatrix} 1 & \alpha & \alpha\beta \\ \alpha & 1 & \beta \\ \alpha\beta & \beta & 1 \end{bmatrix} \tag{3}$$

Note that the assumption of perfect mediation forces the covariance between $x$ and $y$ to be $\alpha\beta$. In any finite data set, however, the observed covariance matrix, $S = [S_{ij}]$, will not typically satisfy this constraint.

The general log-likelihood fitting function for an SEM is given by

$$L = \text{tr}\left(S\Sigma^{-1}\right) + \log|\Sigma|, \tag{4}$$

where $|\cdot|$ denotes the determinant of a matrix and $\text{tr}(\cdot)$ denotes the trace[85]. For the perfect-mediation model, these values are

$$|\Sigma| = (1-\alpha^2)(1-\beta^2) \tag{5}$$

$$\Sigma^{-1} = \begin{bmatrix} 1/(1-\alpha^2) & \alpha/(1-\alpha^2) & 0 \\ \alpha/(1-\alpha^2) & (1-\alpha^2\beta^2)/\left((1-\alpha^2)(1-\beta^2)\right) & \beta/(1-\beta^2) \\ 0 & \beta/(1-\beta^2) & 1/(1-\beta^2) \end{bmatrix} \tag{6}$$

Plugging these into the likelihood function, we get

$$L = \log\left((1-\alpha^2)(1-\beta^2)\right) + \frac{3-\alpha^2-\beta^2-\alpha^2\beta^2}{(1-\alpha^2)(1-\beta^2)} + \frac{2\alpha}{1-\alpha^2}S_{12} + \frac{2\beta}{1-\beta^2}S_{23} \tag{7}$$

To simplify notation, we define

$$F(\alpha, \beta) = \log\left((1-\alpha^2)(1-\beta^2)\right) + \frac{3 - \alpha^2 - \beta^2 - \alpha^2\beta^2}{(1-\alpha^2)(1-\beta^2)}, \qquad (8)$$

so the likelihood function is now

$$L = F(\alpha, \beta) + \frac{2\alpha}{1-\alpha^2} S_{12} + \frac{2\beta}{1-\beta^2} S_{23} \qquad (9)$$

Note that this likelihood is maximized by fitting regression coefficients $\alpha$ and $\beta$ between $x$ and $m$ and $m$ and $y$, respectively, but the log-likelihood formulation is useful for the multivariate extension below.

**Projecting multivariate data to identify latent mediators.** Suppose now that we have three data matrices, $X$, $M$, and $Y$ (individuals by variables) that are mean centered by column. The central assumption of HDMA is that these multivariate data encode latent variables that are causally linked according to the perfect-mediation model, in a sense made precise as follows.

We use the log-likelihood function (Eqn. (7)) of the perfect mediation model as an objective function to identify latent variables, $l_X$, $l_M$, and $l_Y$, that are are correlated as closely as possible to the constraints of the perfect mediation model, Eqn. (3). We estimate these latent variables as linear combinations of the measured variables

$$l_X = Xa \qquad (10)$$

$$l_M = Mb \qquad (11)$$

$$l_Y = Yc \qquad (12)$$

The coefficient vectors $a$, $b$, and $c$, are called loadings, analogous to the terminology in PCA and CCA. Because the data matrices are mean centered, we have

$$\text{mean}(l_X) = \text{mean}(l_M) = \text{mean}(l_Y) = 0, \qquad (13)$$

and we assume the loadings are scaled so that each latent variable has unit variance

$$\text{var}(l_X) = \text{var}(l_M) = \text{var}(l_Y) = 1. \qquad (14)$$

Plugging these formulae into the objective function (Eqn. (9)), we have

$$S_{12} = \text{corr}(l_X, l_M) \qquad (15)$$

$$S_{23} = \text{corr}(l_M, l_Y) \qquad (16)$$

$$L(\alpha, \beta, a, b, c) = F(\alpha, \beta) + \frac{2\alpha}{1-\alpha^2} \text{corr}(l_X, l_M) + \frac{2\beta}{1-\beta^2} \text{corr}(l_M, l_Y) \qquad (17)$$

$$= F(\alpha, \beta) + \frac{2\alpha}{1-\alpha^2} \text{corr}(Xa, Mb) + \frac{2\beta}{1-\beta^2} \text{corr}(Mb, Yc) \qquad (18)$$

This yields an objective function of two sets of parameters: the structural parameters $\alpha$ and $\beta$ that define the causal model among latent variables, and the loading vectors $a$, $b$, and $c$, that define the latent variables in terms of the measured variables. The goal of HDMA is to optimize $L$ as a function of all parameters simultaneously. The form of the objective function, Eqn. (17), is effectively a weighted sum of correlation coefficients, connecting it to so-called sum-of-correlation, or SUMCOR, optimization problems[86], which we discuss further below.

**An algorithm for HDMA.** The global optimization of (17) is challenging because it is not a convex problem. However, the decomposition of the variables into structural and loading variables suggests an iterative algorithm, similar to the expectation-maximization algorithm, that converges at least to a stationary point. The overall idea is to use a block-coordinate-ascent strategy that iterates between optimizing $a$, $b$, and $c$, then optimizing $\alpha$ and $\beta$.

For fixed $a$, $b$, and $c$, the optimal $\alpha$ and $\beta$ are simply given by regression coefficients between $l_X$ and $l_M$ and $l_M$ and $l_Y$, respectively. Given these regression coefficients, $\alpha$ and $\beta$, we then optimize $a$, $b$, and $c$. For fixed $\alpha$ and $\beta$, the term $F(\alpha, \beta)$ is irrelevant, so maximizing the log-likelihood function reduces to maximizing the reduced function

$$L_{red}(a, b, c) = \frac{2\alpha}{1-\alpha^2} \text{corr}(Xa, Mb) + \frac{2\beta}{1-\beta^2} \text{corr}(Mb, Yc), \qquad (19)$$

which is a weighted sum of correlation coefficients. This is exactly a (weighted) SUMCOR optimization problem[86]. These optimization problems are still not convex, but Tenenhaus et al. have recently proved convergence for iterative algorithms that optimize weighted SUMCOR problems[86–88]. These algorithms only guarantee convergence to a stationary point not necessarily a maximum, as is common in other non-convex problems, but this can be overcome with multiple random restarts, if needed. Thus, we have a sub-routine $\text{wSUMCOR}(X, M, Y, w_1, w_2)$ that solves the weighted SUMCOR problem

$$L_{wSUMCOR}(a, b, c, w_1, w_2) = w_1 \text{corr}(Xa, Mb) + w_2 \text{corr}(Mb, Yc). \qquad (20)$$

Iterating between optimizing the structural parameters and loading parameters, we reduce the negative log-likelihood at each step and converge to a fixed point.

We summarize our optimization procedure in Algorithm 1.

**Algorithm 1**. High-dimensional mediation analysis
 **Intput:** $X$, $M$, $Y$ ▷ Data matrices
 **Output:** $\alpha$, $\beta$, $a$, $b$, $c$, $l_X$, $l_M$, $l_Y$ ▷ Structural parameters, loadings, scores
 $\alpha \leftarrow 0.5$, $\beta \leftarrow 0.5$ ▷ Initialize structural parameters
 **While** *converge* ≠ *TRUE* **do**
 $d \leftarrow \frac{2\alpha}{1-\alpha^2} + \frac{2\alpha}{1-\alpha^2}$ ▷ Normalization constant for weights
 $w_1 \leftarrow \frac{1}{d}\frac{2\alpha}{1-\alpha^2}$, $w_2 \leftarrow \frac{1}{d}\frac{2\beta}{1-\beta^2}$ ▷ Set weights (sum to one)
 $(a, b, c) \leftarrow \text{wSUMCOR}(X, M, Y, w_1, w_2)$ ▷ Compute loadings
 $l_X \leftarrow Xa$, $l_M \leftarrow Mb$, $l_Y \leftarrow Yc$ ▷ Compute scores $\alpha \leftarrow \text{corr}(l_X, l_M)$, $\beta \leftarrow \text{corr}(l_M, l_Y)$ ▷ Update structural parameters
 **end while**

**Kernel HDMA.** For large data matrices $X$, $M$, and $Y$, especially with high correlation among variables, as is common for high-throughput biological assays (e.g., ~1M alleles for genotypes, ~20k transcripts), we can further reduce the dimensionality of the HDMA model by requiring that loading vectors lie in the span of the the measured individuals, namely

$$a = X^T \tilde{a} \qquad (21)$$

$$b = M^T \tilde{b} \qquad (22)$$

$$c = Y^T \tilde{c}. \qquad (23)$$

This replaces the full feature data, say $X$, with the covariances among individuals (aka, Gram matrices), $C_X = XX^T$, and reduces the

dimensionality from the number of measured variables down to the number of individuals

$$l_x = XX^T \tilde{a} = C_X \tilde{a} \qquad (24)$$

$$l_M = MM^T \tilde{b} = C_M \tilde{b} \qquad (25)$$

$$l_M = YY^T \tilde{c} = C_Y \tilde{c}. \qquad (26)$$

This reduction is called kernelization[88] and is widely applied to other linear models, including CCA, linear regression, and classification.

It is interesting to note that kernelization is often used to convert a linear model to a non-linear model by replacing the covariance matrices, e.g., $C_X$, with more complex kernel matrices $K_X$ that encode similarity measures among individuals that are non-linear functions of the measured variables. non-linear model by replacing the covariance matrices, e.g., $C_X$, with more complex kernel matrices $K_X$ that encode similarity measures among individuals that are non-linear functions of the measured variables. Promoting a linear model to a non-linear model in this way is called the kernel trick and is widely used in the machine learning field. The above considerations show that HDMA is kernelizable in the same way as other linear models, although the exploration of non-linear models is outside the scope of this study.

We generated kernel matrices for the genome, phenome, and transcriptome as described above. To test the effect of the presence of local eQTLs on mediation, we further generated two additional transcriptomic kernels. 1) A distal-only kernel was derived first by regressing out the effect of local haplotype on all transcripts as described above and generating the kernel matrix using the residual expression (distal-affected only). 2) A local-only kernel was derived by imputing transcription levels for each transcript as described above and then calculating the kernel with only these locally derived expression values. We replaced the original transcriptomic kernel with each of these additional kernels in turn and performed HDMA. We calculated the correlation between all pairs of latent variables and the path coefficient for each instance.

**Implementation details.** We have implemented HDMA (Algorithm 1) in the R programming language. Tenenhaus et al. have implemented their optimizers in the Regularized Generalized Canonical Correlation Analysis (RGCCA) R package[89], which we use as the subroutine `wSUMCOR`. As Tenenhaus et al. discuss optimizing the empirical correlation coefficient per se is numerically unstable due to the inversion of the covariance matrices of the measured variables (e.g., the transcript-transcript covariance matrix). To overcome this, the RGCCA package uses a regularized form of the covariance matrix developed by Schaeffer and Strimmer[90], which can be estimated rapidly using an analytic formula.

As a convergence criterion, we stop the iterations when both $\alpha$ and $\beta$ change by less than $10^{-6}$ from their previous value in one iteration.

All code required to run HDMA is available at Figshare: https://figshare.com/https://doi.org/10.6084/m9.figshare.27066979[79].

**Enrichment of biological terms**
We performed gene set enrichment analysis (GSEA)[40] using the transcript loadings in each tissue as gene weights. GSEA determines enrichment of pathways based on where the contained genes appear in a ranked list of genes. If the genes in the pathway are more concentrated near the top (or the bottom) of the list than expected by chance, the pathway can be interpreted as being enriched with positively (negatively) loaded transcripts. We used the R package fgsea[39] to calculate normalized enrichment scores for all GO terms and all KEGG pathways.

We downloaded all KEGG[91] pathways for *Mus musculus* using the R package clusterProfiler[91]. We then used fgsea to calculate enrichment scores in each tissue using the transcript loadings in each tissue as our ranked list of genes. We reported the normalized enrichment score (NES) for the 10 pathways with the largest positive NES and the 10 pathways with the largest negative NES.

We used the R package pathview[92] to visualize the loadings from each tissue in interesting pathways. We scaled the loadings in each tissue by the maximum absolute value of loadings across all tissues to compare them across tissues.

We downloaded GO term annotations from Mouse Genome Informatics at the Jackson Laboratory[93] https://www.informatics.jax.org/downloads/reports/index.html. We removed gene-annotation pairs labeled with NOT, indicating that these genes were known not to be involved in these GO terms. We also limited our search to GO terms with between 80 and 3000 genes. We used the R package annotate[94] to identify the ontology of each term and the R package pRoloc[95] to convert between GO terms and names. As with the KEGG pathways, we used fgsea to calculate a normalized enrichment score for each GO term and collected loadings for the transcripts in each term to compare across tissues.

**TWAS in DO mice**
We performed a transcriptome-wide analysis (TWAS)[9,11] in the DO mice to compare to the results of high-dimensional mediation. To perform TWAS, we fit a linear model to explain variation in each transcript across the population using the genotype at the nearest marker to the gene transcription start site (TSS). We used kinship as a random effect and sex, diet, and DO generation as fixed effects. The predicted transcript from each of these models was the imputed transcript based only on the local genotype.

We correlated each imputed transcript with each of the metabolic phenotypes after adjusting phenotypes for sex, diet, DO generation, and DO wave. To calculate significance of these correlations, we performed permutation testing by shuffling labels of individual mice and recalculating correlation values. Significant correlations were those more extreme than any of the permuted values, corresponding to an empirical $p$-value of 0. These are transcripts whose locally encoded expression level was significantly correlated with one of the metabolic traits. This suggests an association between the genetically encoded transcript level and the trait but does not identify a direction of causation.

**Literature support for genes**
To determine whether each gene among those with large loadings or large heritability had a supported connection to obesity or diabetes in the literature, we used the R package easyPubMed[96]. We searched for the terms (diabetes OR obesity) along with the tissue name (adipose, islet, liver, or muscle), and the gene name. We restricted the gene name to appear in the title or abstract as some short names appeared coincidentally in contact information. We checked each gene with apparent literature support by hand to verify that support, and we removed spurious associations. For example, FAU is used as an acronym for fatty acid uptake and CAD is used as an acronym for coronary artery disease. Both terms co-occur with the terms diabetes and obesity in a manner independent of the genes *Fau* and *Cad*. Other genes that co-occurred with diabetes and obesity, but not as a functional connection were similarly removed. For example, the gene *Rpl27* is used as a reference gene for quantification of the expression of other genes, and co-occurrence with diabetes and obesity is a coincidence. We counted the abstracts associated with diabetes or obesity and each gene name and determined that a gene had literature support when it had at least two abstracts linking it to the terms diabetes or obesity in the respective tissue.

### Tissue-specific clusters

To compare the top loading genes across tissues, we selected genes with a loading at least 2.5 standard deviations from the mean across all tissues. We made a matrix consisting of the union of these sets populated with the tissue-specific loading for each gene. We used the pam() function in the R package cluster[97] to cluster the loading profiles around $k$ medoids. We tested $k = 2$ through 20 and used silhouette analysis to compare the separation of the clusters. The best separation was achieved with $k = 12$ clusters. For each cluster we used the R package gprofiler2[98] to identify enriched GO terms and KEGG pathways for the genes in each cluster.

### Imputation of gene expression in CC-RIX

To impute gene expression in the CC-RIX, we performed the following steps for each transcript in each tissue (adipose, liver, and skeletal muscle):

1. Calculate diploid CC-RIX genotype for all CC-RIX individuals at the marker nearest the transcription start site of the transcript.
2. Multiply the genotype probabilities by the eQTL coefficients identified in the DO population.

To check the accuracy of the imputation, we correlated each imputed transcript with the measured transcript. The average Pearson correlation (r) was close to 0.5 for all three tissues (Supplementary Fig. 9A), and as expected, the correlation between the imputed transcript and the measured transcript was highly positively dependent on the local eQTL LOD score of the transcript (Supplementary Fig. 9B).

### Prediction of CC-RIX traits

We used both measured expression and imputed expression combined with the results from HDMA in the to predict metabolic disease index (MDI) in the CC-RIX. The traits measured in the DO and the CC-RIX were not identical, so we limited our prediction to body weight, which was measured in both populations, and was the largest contributor to MDI in the DO.

For each CC-RIX individual, we multiplied the transcript abundances across the transcriptome by the loadings derived from the HDMA in the DO population. This resulted in a vector with $n$ elements, where $n$ is the number of transcripts in the trancriptome. Each element was a weighted value that combined the relative abundance of the transcript with how that abundance affected MDI. We averaged the values in this vector to calculate an overall predicted MDI for the individual CC-RIX animal.

After calculating this predicted MDI across all CC-RIX animals, we correlated the predicted values from each tissue with measured body weight (Supplementary Fig. 9B).

### Cell type specificity

We investigated whether the loadings derived from HDMA reflected tissue composition changes in the DO mice prone to obesity on the high-fat diet. To do this, we acquired lists of cell-type specific transcripts from the literature. In adipose tissue, we looked at cell-type specific transcripts for macrophages, leukocytes, adipocyte progenitors, and adipocytes as defined in Ehrlund et al. (2017)[99]. In pancreatic islets, we looked at cell-type specific transcripts for alpha cells, beta cells, delta cells, ductal cells, mast cells, macrophages, acinar cells, stellate cells, gamma and epsilon cells, and endothelial cells as defined by Elgamal et al. (2023)[100]. Both studies defined cell-type specific transcripts based on human cell types. We collected the loadings for each set of cell-type specific transcripts in the respective tissue and asked whether the mean loading for the cell type differed significantly from 0. A significant positive loading for the cell type would suggest a genetic predisposition to have a higher proportion of that cell type in the tissue. To determine whether each mean loading differed significantly from 0, we performed permutation tests. We randomly sampled $n$ genes outside of the cell-type specific, where $n$ was the number of genes in the set. We compared the distribution of loading means over 10,000 random draws to that seen in the observed data. We used a significance threshold of 0.01.

### Comparison of transcriptomic signatures to human transcriptomic signatures

To compare the transcriptomic signatures identified in the DO mice to those seen in human patients, we downloaded human gene expression data from the Gene Expression Omnibus (GEO)[101,102]. We focused on adipose tissue because this had the strongest relationship to obesity and insulin resistance in the DO. We downloaded the following human gene expression data sets:

- Accession number GSE152517 - Bulk RNA sequencing on visceral adipose tissue resected from seven diabetic and seven non-diabetic obese individuals.
- Accession number GSE44000 - Agilent-014850 4X44K human whole genome platform arrays (GPL6480) measuring gene expression in purified adipocytes derived from the subcutaneous adipose tissue of seven obese (BMI>30) and seven lean (BMI<25) post-menopausal women.
- Accession number GSE205668 - Subcutaneous adipose tissue was resected during elective surgery from 35 normal weight, and 26 obese children. Gene expression was measured by RNA sequencing with an Illumina HiSeq 2500.
- Accession number GSE29231 - Visceral adipose biopsies were taken from three (four technical replicates each) female patients with type 2 diabetes, and three (four technical replicates each) non-diabetic female patients. Expression was measured with Illumina HumanHT-12 v3 Expression BeadChip arrays.

We downloaded each data set from GEO using the R package GEOquery[103]. In each case, we verified that gene expression was log transformed and performed the transformation ourselves if it had not already been done. When covariates such as age and sex were available in the metadata files, we regressed out these variables (GSE205668— sex and age; GSE44000—none; GSE29231—age; GSE152517—none). We mean-centered and standardized gene expression across transcripts.

We matched the human gene expression to the mouse gene expression by pairing orthologs as defined in The Jackson Laboratory's mouse genome informatics data base (MGI)[104]. We multiplied each transcript in the human data by the adipose tissue loading of its ortholog in the DO mice. This resulted in a vector of weighted transcript values for each patient based on their own transcriptional profile and the obesity-related transcriptional signature from the DO analysis. The mean of this vector for an individual was the prediction of their obesity status (MDI). Higher values indicate a prediction of higher obesity or risk of metabolic disease based on adipose gene expression.

We then compared the values across groups, either obese and non-obese, or diabetic and non-diabetic depending on the groups in each study. For the three studies without technical replicates (accession numbers GSE152517, GSE44000, and GSE205668), we used Welch's t-test to compare the means of the two groups. This is the default t-test in base R and assumes unequal variance across groups. For the study that included technical replicates (GSE29231) we fit a linear mixed model from the R packages lme4[82] and lmerTest[105]. We used subject ID as a random variable.

### Connectivity map queries

We queried the transcript loading signatures from adipose tissue and pancreatic islets with the CMAP database. These tissues are the most related to metabolic disease and diabetes respectively.

The gene expression profiles in the Connectivity Map database are derived from human cell lines and human primary cultures and are indexed by Entrez gene IDs. To query the CMAP database, we identified the Entrez gene IDs for the human orthologs of the mouse genes

expressed in each tissue. Each CMAP query takes the 150 most up-regulated and the 150 most down-regulated genes in a signature, how-ever, not all human genes are included in their database. To ensure we had as many genes as possible in the query, we selected the top and bottom 200 genes with the most extreme positive and negative loadings respectively. We pasted these into the CLUE query application available at https://clue.io/query. These gene lists are available as Source Data.

We filtered the results in two ways: First, we looked at the most significantly anti-correlated ($-\log_{10}(\text{FDR }q) > 15$) hits across all cell types. Second, we looked at the most anti-correlated within the most related cell type to the query and considered hits regardless of $-\log_{10}(\text{FDR }q)$. For adipose tissue we looked in normal adipocytes, abbreviated ASC in the CMAP database, and for pancreatic islets we looked in pancreatic cancer cells, abbreviated YAPC in the CMAP database.

## Reporting summary
Further information on research design is available in the Nature Portfolio Reporting Summary linked to this article.

## Data availability
**DO mice:** Genotypes, phenotypes, and pancreatic islet gene expression data were previously published[12]. Gene expression for the other tissues can be found at the Gene Expression Omnibus with the following accession numbers: DO adipose tissue - GSE266549; DO liver tissue - GSE266569; DO skeletal muscle - GSE266567. Expression data with cal-culated eQTLs are available at Figshare. 10.6084/m9.figshare.27066979[77]. **CC-RIX mice:** Gene expression can be found at the Gene Expression Omnibus with the following accession numbers: CC-RIX adi-pose tissue - GSE237737; CC-RIX liver tissue - GSE237743; CC-RIX skeletal muscle - GSE237747. Count matrices and phenotype data can be found on Figshare. 10.6084/m9.figshare.27066979[77]. Source Data for figures in this manuscript are available on Figshare. 10.6084/m9.figshare.29247428[106]. **Data from previous publications:** We downloaded gen-otypes, phenotypes, and pancreatic islet gene expression data from Dryadhttps://doi.org/10.5061/dryad.pj105[79]. To predict human MDI from RNA-seq profiles, the following data were downloaded from the Gene Expression Omnibus: * Accession number GSE152517 Gene expression in visceral adipose tissue resected from seven diabetic and seven non-diabetic obese individuals. * Accession number GSE44000 Gene expression in purified adipocytes derived from the subcutaneous adipose tissue of seven obese (BMI > 30) and seven lean (BMI < 25) post-menopausal women. * Accession number GSE205668 Gene expression in subcutaneous adipose tissue from 35 normal weight, and 26 obese children. * Accession number GSE29231 Visceral adipose biopsies were measured in four technical replicates each in three female patients with type 2 diabetes, and three non-diabetic female patients.

## Code availability
**Code**: All code used to run the analyses reported here are available at Figshare: https://figshare.com10.6084/m9.figshare.27066979[77].

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

## Acknowledgements
This project was supported by The Jackson Laboratory Cube Initiative, as well as grants from the National Institutes of Health (grant numbers R01DK101573, R01DK102948, and RC2DK125961 to A. D. A., R01GM115518 to G. W. C., and R01GM141309 to J.M.M- and by the University of Wisconsin-Madison, Department of Biochemistry and Office of the Vice Chancellor for Research and Graduate Education with funding from the Wisconsin Alumni Research Foundation (to M. P. K.). We thank the following scientific services at The Jackson Laboratory: Genome Technologies for the RNA sequencing, necropsy services for the tissue harvests, and the Center for Biometric Analysis for metabolic phenotyping.

## Author contributions
Conceptualization: J.M.M., A.D.A., N.A.R., MJB, G.A.C., G.W.C. Methodology: G.A.C., A.D.A., J.M.M., C.N.B., M.P.K. Software: A.L.T., J.M.M., I.G.G., A.S. Formal Analysis: A.L.T., I.G.G., A.S. Investigation: C.N.B., M.G. Resources: A.D.A., M.P.K., C.N.B., NAR, MJB, G.A.C., G.W.C. Data Curation: C.N.B., A.L.T. Writing - Original Draft: A.L.T, J.M.M. Writing - Review and Editing: A.L.T., J.M.M, G.A.C, G.W.C., M.P.K., C.N.B. Visualization: A.L.T. Supervision: M.J.B., N.A.R., A.D.A., G.A.C., G.W.C. Project Administration: M.J.B., C.N.B. Funding acquisition: N.A.R., A.D.A., G.A.C., G.W.C.

## Competing interests
The authors declare no competing interests.
