## [Transparent Peer Review file · Nature Communications]

Transcripts with high distal heritability mediate genetic effects on complex metabolic traits

Corresponding Author: Dr Gregory Carter

Version 0:

Reviewer comments:

Reviewer #1

(Remarks to the Author)

Tyler et al. have used multiple metabolic phenotypes, genotypes and transcriptome data from four metabolically active tissues from the diversity outbred mice to assess the role of distal vs local gene expression on a composite metabolic phenotype (mainly comprising body weight and insulin sensitivity). They initially found that local eQTL negatively correlated with trait relevance, while distal eQTL positively associated with trait relevance. They then used high dimensional mediation analysis (HDMA) to identify transcriptome signatures that are heritable, correlate with the composite metabolic phenotype, and impact the phenotype in a casual fashion (e.g., when taken into account, the signal between genotype and phenotype disappears). They found that transcripts that contributed most strongly to the phenotype were under distal, as opposed to local genetic control and were able to identify relevant biological pathways. They then used the transcript loadings to validate these findings in CC-RIX mice and human data. The manuscript convincingly demonstrates the importance of distal eQTL for metabolic phenotypes, a finding supported by recent publications. By comparing to TWAS and genes with high local heritability, this work emphasizes the complexity of the genetic architecture underlying complex traits and argues against a focus on local eQTL. The use of HDMA is a strength and represents a tool that will likely be used by others for other models and phenotypes. The work does not highlight any new biological pathways - everything they found has previously been reported in the literature. It is not surprising that transcriptomic signatures involved in inflammation or mitochondrial function play a role in obesity, and therefore one would expect these to be replicated in the CC-RIX and human cohorts. Demonstrating tissue-specificity is also not new. Despite lack of novel biological pathways, this work clearly demonstrates the ability of HDMA to identify appropriate transcriptomic signatures as well as the importance of distal eQTL for complex traits.

Minor comments:

- 1) In the results section, Fig. 3a, is the composite transcriptome score for all tissues combined? Would partial correlations differ if tissues were looked at individually?
- 2) Results: In HDMA, transcripts are given loadings and these loadings are used to perform GSEA. This allows you to determine which biological pathways are altered in the composite phenotype. You do not mention the number of genes that have high or low loadings for each tissue. In other words, how many genes fall into the transcriptomic signatures that satisfy HDMA criteria (heritable, correlate with trait and fits a causal model). It would be worthwhile to include this information as a supplementary table.
- 3) Methods: CC-RIX, you mention that a sub-set of mice were treated with metformin, but there is no mention of how these data are used in the results. Similarly, CC-RIX mice were euthanized at two time-points (6 months and 12 months) – what time-point was used in the current analysis?
- 4) Methods: you state that pancreas was used for whole pancreas insulin content. If this is the case, was a different data-set used to isolate islets for RNAseq? There is also no mention of islets under the RNAseq methods section.
- 5) Brain is extremely important tissue for metabolic traits such as obesity. Data here demonstrate that adipose tissue is the most relevant for obesity from the four tissues that were looked at, but brain is likely to have more relevance than adipose tissue. This may be worth a mention.
- 6) Line 148: CCA is not defined

(Remarks on code availability)

Reviewer #2

(Remarks to the Author)

Tyler et al. conducted an interesting study using a population of Diversity Outbred mice, uncovering a stronger association between distal genetic elements and traits. This partly explains why local genetic variants contribute less to traits via transcripts. The findings also support a previously observed phenomenon in human genetics: “transcripts with low local heritability explain more expression-mediated disease heritability than transcripts with high local heritability.” Below are some comments:

- “Overall, local and distal genetic factors contributed approximately equally to transcript abundance”: Is this similar in human genetics? Do the GTEx data support this conclusion?
- If I understand correctly, in the high-dimensional mediation analysis, the composite transcriptome score includes a non-genetic component. Could the framework capture the condition that transcripts are influenced by traits?
- The authors used the cMap data for the analysis. Are these cell-level data technically replicated? If so, are the technical replicate results consistent?
- “To assess the importance of genetic regulation of transcript levels to clinical traits, we compared the local and distal heritabilities of transcripts to their trait relevance, defined as the maximum trait correlation for each transcript.” Was transcript level standardized during this process? How exactly is this “maximum trait correlation” defined?

Minor Comments:

1. Please define “local” and “distal” prominently.
2. In line 110: “We calculated the heritability of each transcript” — the word “calculate” should be replaced with “estimate.”
3. In line 148, when CCA first appears, please provide its full name.

(Remarks on code availability)

Reviewer #3

(Remarks to the Author)

This study uses a cohort of diversity outcross mice to perform a novel mediation analysis of obesity-related traits. The two main findings are that (i) distal eQTLs (ie trans eQTLs) seem to play a larger role in controlling transcripts relevant to obesity traits than do local (cis) eQTLs, and (ii) it is possible to define composite measures of genotype and transcriptome and phenotype and perform a mediation analysis on the composite measures, thereby establishing causality.

Both of these findings are interesting and noteworthy and would be of interest to the readership of Nature Comms. However, I also had a large number of queries about the manuscript. Whilst most of these are related to improving the presentation, there are a two more substantive queries which need to be dealt with satisfactorily.

Major points:

(i) Is the permutation procedure (Fig 3B,C) valid? It ignores relationships between individuals across omic levels (ie assumes all mice are exchangeable) and so might inflate the apparent significance. If I have understood correctly, the permutation procedure is performed such that it destroys any correlation between the three omic levels – including the existence of any QTLs or eQTLs, and this is likely too harsh a null hypothesis. It might be better to use multivariate generative models to simulate sets of genotypes, transcripts and phenotypes and evaluate performance on those rather than using permutation. An alternative might be to ask how unusual are the correlations $\text{cor}(P_C, T_C)$ etc for the optimal choices of weights compared to randomly sampled weights using the unpermuted data, possibly augmented with a distribution fitted to the random correlations. In any event, that authors should justify their choice of permutation strategy and explain why it supports their thesis.

(ii) I'm a bit puzzled by the use of the CC-RIX mice as a validation set. It appears that actual body weights were not measured in the CC-RIX (why not? – weight is a standard phenotype) so they were imputed from local transcript data, which complicates their use for validation and makes it far less convincing. I really don't see what they add to the study. Why not simply keep back a random 10% of the DO mice, train the models on the 90% and test the predictions into the 10%, for many random samplings? (ie the standard machine learning cross validation procedure).

Minor Points:

(i) Figure 1 is very good, except Figure 1G could be improved if the upper triangle of the heatmap displayed genetic correlations and the main diagonal heritabilities.

(ii) Figure 2 could be significantly improved by replacing the violin plots in Fig2A with overlapping distributions as in Figure 1A,B. Fig 2B is informative but I think the use of linear regression is not the best way of showing the shapes of the distributions for Local and Distal are different. Clearly a straight line does not fit any of the data very well. Can the authors think of another measure which quantifies the fact that strong local eQTL are more likely to have small trait correlations than

strong distal eQTL?

(iii) Fig 3 is hard to follow, particularly since it mentions Kernelization which is otherwise not mentioned in the main paper (it's mentioned in the Methods) Not sure what G_K, T_K, P_K signify.

(iv) Figure 4 D – Please include scatter plots of P_C vs T_C, G_C vs T_C, G_C vs P_C - this is surely key to understanding. Please replace violin plots with overlapping distributions as in Figure 1A,B

(v) Fig 5 is good. It would be helpful to define exactly what is meant by TWAS in this study. I suggest a completely different color ramp is used to indicate tissue type from that used to indicate heritability – it's a bit confusing. It would also help to report the p-values of the t-tests for comparing the heritability distributions for distal vs local (in the three inset boxplots in the Fig 5)

(vi) Given the diverse ancestry of the DO, involving alleles from three different murine subspecies, it would be interesting to know if the distal or local eQTLs more often involved alleles segregating between subspecies.

(vii) Is there a reason for preferring the nomenclature “Local vs Distal” instead of the more usual “cis vs trans”?

(viii) The statements around line 280: The mean loadings for alpha-cell specific transcripts were significantly greater than 0, while the mean loadings for delta- and endothelial-cell specific genes were significantly less than 0 (Fig. 8B) The study does not give p-values for these statements – Fig 8B appears to show boxplots which don't give an indication of significance. There are several places in the MS where boxplots are used as evidence of “significance” without a formal p-value being provided.

(ix) What were the exact criteria for calling eQTL (local and distal) and pQTL? Were different significance thresholds applied for local vs distal? The methods are vague – why does a LOD score threshold of 8 equate to a p-value of 0.05 (and what does this p-value mean – it is genome wide significance?) Surely it would be better to use an FDR-based threshold.

(x) Why was heritability computed from called eQTLs rather than from suitably partitioned genetic relationship matrices?

(xi) The human validation material is quite interesting but I am not an expert on this part of the paper. It did seem overly long.

(xii) The discussion is incredibly short – there was no attempt to place the findings in a wider context.

Methods:

(i) Genotyping (line 25 onwards) Why were haplotypes determined from RNAseq reads and not from the SNP genotypes?

(ii) It not clear whether the CC-RIX mice were kept in the same animal facility as the DO mice or were from a different experiment. Were they on the same high-fat diet as the DO? - please clarify.

(iii) It is not clear what the “processed data” (Methods line 31) refer to. Are these the CC-RIX genotypes? If so, what are the gene expression data?

(iv) TWAS analysis (methods line 250 onwards. Using just the SNP closest to the TSS for each gene might result in underestimating local genetic effects – it would have been better to have taken the most associated SNP within say 100kb. Not all cis SNPs will be associated with the expression trait, so picking one based solely on location is sub-optimal.

(Remarks on code availability)

The link to the code <https://figshare.com/DOI:10.6084/m9.figshare.27066979228> does not open

Version 1:

Reviewer comments:

Reviewer #1

(Remarks to the Author)

The authors have addressed my previous concerns. I have no further concerns.

(Remarks on code availability)

Reviewer #2

(Remarks to the Author)

The author's response has dispelled my concerns, and I have no further questions.

(Remarks on code availability)

Reviewer #3

(Remarks to the Author)

The authors have dealt satisfactorily with most of my comments, but there some outstanding points.

I have added my detailed responses in red to the rebuttal letter, which is probably the clearest way to convey them because the original points were not numbered.

In summary:

Thanks for clarifying the permutation procedure. I don't think it is really addressing the question of whether T_C is unusually well correlated with P_C. However, I can see that it would be very difficult to come up with a solution, so provided the authors make very clear the limitations of their permutation procedure I guess that is OK.

I still think there is a case for a training/test set analysis within the DO, in addition to the RIX, because of the challenges interpreting the replication experiments due to differences in diet, husbandry etc between the two populations. That said, it is quite impressive that the RIX replicates the DO despite all these differences and using RIX body weight as a surrogate phenotype.

The MS should make it crystal clear it is using a 5% threshold for eQTLs and preferably use a more stringent threshold.

I also don't think the violin plots add much - they are not the best way for showing differences between very similar distributions but this is a matter of personal preference.

(Remarks on code availability)

Version 2:

Reviewer comments:

Reviewer #3

(Remarks to the Author)

The second revision satisfactorily addresses all my queries. This is a nice study.

(Remarks on code availability)

Dear Editors and Reviewers,

Thank you so much for your careful reading of our manuscript. We appreciate your time and the feedback you have provided. We have addressed each of your comments, which has greatly improved the clarity of the manuscript.

Changes to the manuscript include the following:

- 1) Multiple additions of clarifying language in the main text and the methods section.
- 2) Color changes for the tissues to avoid confusion with heat maps.
- 3) Shortening of results section relating to CMAP analysis.
- 4) Alteration of Figure 2B and C to use non-parametric statistics and lines
- 5) Alteration of Figure 4D to show null distributions and numbers of transcripts with large loadings.
- 6) Verification that all numeric results in the manuscript were programmatically reported rather than entered by hand. Some numbers have changed slightly but did not affect any conclusions.

Individual responses to comments are below in indented text. We have also provided manuscript documents with tracked changes for ease of comparison.

Thank you again for your efforts.

Sincerely,
Gregory W. Carter

Reviewer #1 (Remarks to the Author):

Tyler et al. have used multiple metabolic phenotypes, genotypes and transcriptome data from four metabolically active tissues from the diversity outbred mice to assess the role of distal vs local gene expression on a composite metabolic phenotype (mainly comprising body weight and insulin sensitivity). They initially found that local eQTL negatively correlated with trait relevance, while distal eQTL positively associated with trait relevance. They then used high dimensional mediation analysis (HDMA) to identify transcriptome signatures that are heritable, correlate with the composite metabolic phenotype, and impact the phenotype in a casual fashion (e.g., when taken into account, the signal between genotype and phenotype disappears). They found that transcripts that contributed most strongly to the phenotype were under distal, as opposed to local genetic control and were able to identify relevant biological pathways. They then used the transcript loadings to validate these findings in CC-RIX mice and human data. The manuscript convincingly demonstrates the importance of distal eQTL for metabolic phenotypes, a finding supported by recent publications. By comparing to TWAS and genes with high local heritability, this work emphasizes the complexity of the genetic architecture underlying complex traits and argues against a focus on local eQTL. The use of HDMA is a strength and represents a tool that will likely be used by others for other models and phenotypes. The work does not highlight any new biological pathways - everything they found has previously been reported in the literature. It is not

surprising that transcriptomic signatures involved in inflammation or mitochondrial function play a role in obesity, and therefore one would expect these to be replicated in the CC-RIX and human cohorts. Demonstrating tissue-specificity is also not new. Despite lack of novel biological pathways, this work clearly demonstrates the ability of HDMA to identify appropriate transcriptomic signatures as well as the importance of distal eQTL for complex traits.

Minor comments:

1) In the results section, Fig. 3a, is the composite transcriptome score for all tissues combined? Would partial correlations differ if tissues were looked at individually?

Yes, the numbers presented in Fig. 3a are for all tissues combined. We also performed high dimensional mediation on the tissues separately to compare the results. We compared the variance explained by the model as well as the correlations between the latent variables (figures below). Combining the tissues explained slightly more variance than using the tissues separately (upper figure), and the correlations between the latent variables were comparable (lower figure). Because we aimed to determine the maximum variance that could be explained, we used the tissues together. Since the addition of this information to the manuscript may be more confusing than helpful, we present these results only in this response.

2) Results: In HDMA, transcripts are given loadings and these loadings are used to perform GSEA. This allows you to determine which biological pathways are altered in the composite phenotype. You do not mention the number of genes that have high or low loadings for each tissue. In other words, how many genes fall into the transcriptomic signatures that satisfy HDMA criteria (heritable, correlate with trait and fits a causal model). It would be worthwhile to include this information as a supplementary table.

We agree that this information is important. To address this, we compared the observed distributions to null transcript loading distributions. The observed distributions extended well beyond the tails of the null distributions. We have modified Figure 4 to show the transcript loadings for each tissue separately compared to a null distribution. We have marked the number of genes that have loadings beyond the tails of the null distribution for each tissue. The new Figure 4D is reproduced below for reference.

3) Methods: CC-RIX, you mention that a sub-set of mice were treated with metformin, but there is no mention of how these data are used in the results. Similarly, CC-RIX mice were euthanized at two time-points (6 months and 12 months) – what time-point was used in the current analysis?

Thank you for noticing this error! The cohort that was given metformin will be published in later analyses but was not included in this study. We have deleted the sentences referring to metformin. Both the 6- and 12-month cohorts were used in the CC-RIX analysis. We used age as a covariate in all analyses, and we have added language to the methods to indicate this (Lines 51-53 in the Methods with tracked changes).

4) Methods: you state that pancreas was used for whole pancreas insulin content. If this is the case, was a different data-set used to isolate islets for RNAseq? There is also no mention of islets under the RNAseq methods section.

Thank you for catching this omission. The methods regarding the pancreatic islets are described more in detail in the original publication describing these data (PMID: 29567659, ref 12). We have added a short paragraph to the DO section of the methods to describe these measurements (Lines 13-16 in the Methods with tracked changes). The islets in the DO mice were isolated and insulin per islet was measured. The whole pancreas insulin content was calculated from insulin per islet and the total number of islets in each pancreas. RNA was then isolated from the islets for RNA-Seq.

5) Brain is extremely important tissue for metabolic traits such as obesity. Data here demonstrate that adipose tissue is the most relevant for obesity from the four tissues that were looked at, but brain is likely to have more relevance than adipose tissue. This may be worth a mention.

We agree that brain is likely to have gene expression variation relevant to obesity. We have added a sentence to the results to remind readers that the brain is an important tissue in regulation of obesity but was not included in this study (Lines 251-253 in manuscript with tracked changes). Thus, we cannot speak to the relative importance of this tissue in this context.

6) Line 148: CCA is not defined

Thank you for catching this. We have added the definition before the first mention of the term (Line 170 in the manuscript with tracked changes).

Reviewer #2 (Remarks to the Author):

Tyler et al. conducted an interesting study using a population of Diversity Outbred mice, uncovering a stronger association between distal genetic elements and traits. This partly explains why local genetic variants contribute less to traits via transcripts. The findings also support a previously observed phenomenon in human genetics: “transcripts with low local heritability explain more expression-mediated disease heritability than transcripts with high local heritability.” Below are some comments:

- “Overall, local and distal genetic factors contributed approximately equally to transcript abundance”: Is this similar in human genetics? Do the GTEx data support this conclusion?

We have now done a more thorough search of the literature to investigate this question. Multiple human studies have found that the ratio of heritability explained by local and distal regulation is approximately 1:4. In contrast, we found a ratio of about 1:1, that is local and distal genetic factors explained roughly equal amounts of heritability. We suspect that we found a higher contribution of local genotype for several reasons. First, there is a greater divergence between the founder strains of the DO than observed in human populations. Thus, there may be an accumulation of variants that affect local transcription within a given haplotype. Distal effects on transcription, however, are more likely to be buffered by the overall transcriptional network, weakening their observed effects. Second, there is a high degree of linkage disequilibrium in the DO mice compared to human populations, as well as high degree of confidence with which we can estimate ancestral haplotypes. Any given genetic marker in the DO mice captures haplotype information from a relatively large genomic region, thus encompassing many possible nearby regulatory variants. In humans, the information SNPs capture is much more localized and more difficult to assign to ancestral haplotypes. Thus, in mice there may be more information about local regulatory variants captured in the haplotype and therefore more local heritability explained. We have added text to the results section where these data are presented to speculate on the discrepancy between the findings in mice and humans. (Lines 118-127 in manuscript with tracked changes.)

- If I understand correctly, in the high-dimensional mediation analysis, the composite transcriptome score includes a non-genetic component. Could the framework capture the condition that transcripts are influenced by traits?

Yes, absolutely. The results presented here are consistent with a mediation model, but do not prove mediation. We used the results from the CMAP investigation as supportive of the mediation model. That is, drugs that act on the transcriptional pattern also act on the traits, thereby supporting a causal role of the transcriptional profile in the pathogenesis of metabolic disease. This point was not made clear in the original manuscript. We have added text to the discussion to say more explicitly that HDMA alone does not prove mediation but is only consistent with it. We further explain that the results from CMAP help support the mediation model (Lines 384-389 in manuscript with tracked changes).

- The authors used the cMap data for the analysis. Are these cell-level data technically replicated? If so, are the technical replicate results consistent?

The CMAP data include technical replicates with a high degree of consistency across replicates (PMID: 29195078, ref 60 in the manuscript). The results that we present are from the CMAP GSEA query, which takes into account shared modes of action and targeted pathways across individual drugs and cell types. These aggregated statistics take into account fluctuations in drugs in each class as well as across replicates.

- “To assess the importance of genetic regulation of transcript levels to clinical traits, we compared the local and distal heritabilities of transcripts to their trait relevance, defined as the maximum trait correlation for each transcript.” Was transcript level standardized during this process? How exactly is this “maximum trait correlation” defined?

Thank you for pointing out this lack of clarity. We regressed sex, DO wave, and DO generation out of the rank normalized transcript values. We also regressed these covariates out of the phenotypes. We then performed Spearman rank correlation between all pairs of traits and transcripts. The trait relevance of a given transcript was the maximum absolute value of its correlation coefficients across all traits. We have clarified this point in both in the Methods (lines 149-153 Methods with tracked changes) and the main text (lines 130-132 in the manuscript with tracked changes).

Minor Comments:

1. Please define “local” and “distal” prominently.

We have added text to the introduction to define local and distal and to distinguish these terms from cis and trans (Lines 33-37 in manuscript with tracked changes).

2. In line 110: “We calculated the heritability of each transcript” — the word “calculate” should be replaced with “estimate.”

We have changed "calculate" to "estimate" (Line 114 in manuscript with tracked changes).

3. In line 148, when CCA first appears, please provide its full name.

Thank you for catching this. We have added "canonical correlation analysis" to the first instance of CCA (Line 170 of manuscript with tracked changes).

Reviewer #3 (Remarks to the Author):

This study uses a cohort of diversity outcross mice to perform a novel mediation analysis of obesity-related traits. The two main findings are that (i) distal eQTLs (ie trans eQTLs) seem to play a larger role in controlling transcripts relevant to obesity traits than do local (cis) eQTLs, and (ii) it is possible to define composite measures of genotype

and transcriptome and phenotype and perform a mediation analysis on the composite measures, thereby establishing causality.

Both of these findings are interesting and noteworthy and would be of interest to the readership of Nature Comms. However, I also had a large number of queries about the manuscript. Whilst most of these are related to improving the presentation, there are a two more substantive queries which need to be dealt with satisfactorily.

Major points:

(i) Is the permutation procedure (Fig 3B,C) valid? It ignores relationships between individuals across omic levels (ie assumes all mice are exchangeable) and so might inflate the apparent significance. If I have understood correctly, the permutation procedure is performed such that it destroys any correlation between the three omic levels – including the existence of any QTLs or eQTLs, and this is likely too harsh a null hypothesis. It might be better to use multivariate generative models to simulate sets of genotypes, transcripts and phenotypes and evaluate performance on those rather than using permutation. An alternative might be to ask how unusual are the correlations $\text{cor}(P_C, T_C)$ etc for the optimal choices of weights compared to randomly sampled weights using the unpermuted data, possibly augmented with a distribution fitted to the random correlations. In any event, that authors should justify their choice of permutation strategy and explain why it supports their thesis.

We apologize for the lack of clarity on this critical point. The permutation test was performed by only permuting the transcriptomes. The correlations between genome and phenome were preserved in this test. Thus, we preserved the effects of any QTLs while breaking the association between eQTLs (and all other genetic effect on transcription) with QTLs. The essential goal of this procedure was to determine whether it was possible within a random transcriptome to identify a spurious transcriptomic profile that appeared to mediate the true genotype-phenotype correlations as well as our optimized transcriptomic profile in the true data. Our permutation test definitively rules this possibility out; the true transcriptome is more highly aligned with the genotype-phenotype map than expected at random. We stress that this is nontrivial, given that canonical correlation analysis (CCA)-based approaches have problems overfitting because they involved estimating the inverse of a covariance matrix and, hence, can detect such spurious components. We have added clarification of this point, which can be seen in lines 176-179 in the manuscript with tracked changes.

The suggestion to consider the performance of HDMA under generative model hypotheses is an important one, and is the subject of ongoing research, but is beyond the scope of this paper. In this study, we use HDMA in a purely descriptive mode.

We randomly sampled the unpermuted weights 10,000 times and recalculated the correlations $\text{cor}(G_C, T_C)$ and $\text{cor}(T_C, P_C)$ as shown in Figure 3C. The

results are shown below. We think it would add more confusion than information to add this to the manuscript figure, but shows clearly that the correlations identified by HDMA are well outside the null distribution. The original permuted values from the manuscript in Figure 3 are shown in gray, and the observed correlations are shown in red.

(ii) I'm a bit puzzled by the use of the CC-RIX mice as a validation set. It appears that actual body weights were not measured in the CC-RIX (why not? – weight is a standard phenotype) so they were imputed from local transcript data, which complicates their use for validation and makes it far less convincing. I really don't see what they add to the study. Why not simply keep back a random 10% of the DO mice, train the models on the 90% and test the predictions into the 10%, for many random samplings? (ie the standard machine learning cross validation procedure).

We apologize for the lack of clarity on this point. Body weight was measured in the CC-RIX. We used measured body weight as the ground truth validation in the CC-RIX population. We estimated the metabolic disease index (MDI), which is largely based on body weight from the DO and compared it to the actual body weight as the validation (Figure 7B). To estimate MDI, we either used the measured transcriptome in the CC-RIX, or a predicted transcriptome based on local genotype. While the MDI based on the measured transcriptome correlated well with body weight, the MDI based on the locally imputed transcriptome was not correlated with body weight. We interpreted this result as support that genetic effects on genotype are not mediated through local gene regulation. It was important to use the CC-RIX data as opposed to validating within the DO data because the point of local and distal regulation is particularly critical when translating results between populations with different allele structure. We suggest that there is a failure of human TWAS results to translate between human populations because genetic effects are mediated through distal gene regulation, which are dramatically different across populations with different allele structures.

In this mouse experiment, the two populations shared ancestral haplotypes, but had dramatically different allele structure, thus allowing us to tease apart the effects of local and distal gene regulation on phenotypic effects.

Minor Points:

(i) Figure 1 is very good, except Figure 1G could be improved if the upper triangle of the heatmap displayed genetic correlations and the main diagonal heritabilities.

This is an interesting way to add more information to the plot. We tried it (shown below), but then realized that it might be confusing to have two different types of data in a single heat map. Because most of the cells show Pearson correlation, but the diagonal has very low values, we think this might cause more confusion, especially since the heritabilities are shown in Figure 1F.

(ii) Figure 2 could be significantly improved by replacing the violin plots in Fig2A with overlapping distributions as in Figure 1A,B. Fig 2B is informative but I think the use of linear regression is not the best way of showing the shapes of the distributions for Local and Distal are different. Clearly a straight line does not fit any of the data very well. Can the authors think of another measure which quantifies the fact that strong local eQTL are more likely to have small trait correlations than strong distal eQTL?

We tried using overlapping distributions (below), but it became difficult to compare the distributions across the tissues. We would like Figure 2A to show a direct comparison of the variance explained by local and distal eQTLs across tissues, so we reverted to the violin plots.

Figures 2B and C have been changed to avoid the linear test. We binned the transcripts into centiles based on their variance explained. We then calculated the mean and 95th and 5th percentiles of their maximum trait correlation. We smoothed these values using splines. The top and bottom lines in each panel shows the 95th and 5th percentile respectively and the middle line shows the mean. We reported statistics from Spearman rank correlation tests.

(iii) Fig 3 is hard to follow, particularly since it mentions Kernelization which is otherwise not mentioned in the main paper (it's mentioned in the Methods) Not sure what G_K, T_K, P_K signify.

We have added a short description of the kernelization terminology to the main text where we describe high-dimensional mediation. The new text appears in Lines 150-156 of the manuscript with tracked changes.

(iv) Figure 4 D – Please include scatter plots of P_C vs T_C, G_C vs T_C, G_C vs P_C - this is surely key to understanding. Please replace violin plots with overlapping distributions as in Figure 1A,B

We tried using overlapping histograms to compare these distributions, but the distributions are so similar that the figure was difficult to read (see below). Instead, we replaced the distributions with separate panels showing the observed loading distribution for each tissue compared with the null. We shaded the area of the distributions that were more extreme than the null distribution and noted the number of genes in each extreme group. This adds a little more information to help distinguish the distributions.

The scatter plots took up a lot of room in Figure 3 without adding much information, so we have created a new Supplemental Figure S3 to show the details of these correlations.

(v) Fig 5 is good. It would be helpful to define exactly what is meant by TWAS in this study. I suggest a completely different color ramp is used to indicate tissue type from that used to indicate heritability – it's a bit confusing. It would also help to report the p-values of the t-tests for comparing the heritability distributions for distal vs local (in the three inset boxplots in the Fig 5)

We have added a short description of our TWAS procedure to the results section that discusses Figure 5. We have also changed the colors for the tissues to select colors that do not overlap with the heat maps or the local/distal color scheme. We added t-test p values to the legend of Figure 5 for each of the box plots (page 14 in manuscript with tracked changes).

(vi) Given the diverse ancestry of the DO, involving alleles from three different murine subspecies, it would be interesting to know if the distal or local eQTLs more often involved alleles segregating between subspecies.

This is an interesting question and has been addressed extensively in the literature (PMIDs 25230953, 21406540, 34849860, 34425882, 33264334, 31961859). Because eQTLs were only an incidental piece of this manuscript (presented only in supplemental figures) and because this issue has been

examined previously, we performed a limited analysis that is presented only in this response letter. We collected local and distal allele coefficients for all haplotypes. A comparison of the local and distal eQTL coefficients shows that the *castaneus* (green) and *musculus* (red) haplotypes tend to have larger local coefficients than the *domesticus* strains (gray). An ANOVA with Tukey's Honestly Significant Difference showed that the *castaneus*, *musculus*, and the wild-derived *domesticus* strain (WSB) had significantly stronger local eQTL coefficients than the other *domesticus* strains (top figures). For distal coefficients (bottom figures), both *castaneus*, and *musculus* strains had significantly higher coefficients than all *domesticus* strains.

(vii) Is there a reason for preferring the nomenclature “Local vs Distal” instead of the more usual “cis vs trans”?

We avoid the usage of cis and trans terminology because these terms have specific biochemical definitions that are not completely captured by physical location on the genome (See Box 2 in PMID: 18597885, ref 22 in the manuscript). Because we are not evaluating the biochemical nature of variants in this study and are classifying variants only by genomic position, we use the terms local and distal. We have added text to the introduction to define these terms and discuss the difference between cis/trans and local/distal terminology.

(viii) The statements around line 280: The mean loadings for alpha-cell specific transcripts were significantly greater than 0, while the mean loadings for delta- and endothelial-cell specific genes were significantly less than 0 (Fig. 8B)

The study does not give p-values for these statements – Fig 8B appears to show boxplots which don’t give an indication of significance. There are several places in the MS where boxplots are used as evidence of “significance” without a formal p-value being provided.

We apologize for this omission. We have added p values to the text for each of these significance claims.

(ix) What were the exact criteria for calling eQTL (local and distal) and pQTL? Were different significance thresholds applied for local vs distal? The methods are vague – why does a LOD score threshold of 8 equate to a p-value of 0.05 (and what does this p-value mean – it is genome wide significance?) Surely it would be better to use an FDR-based threshold.

Local eQTLs were defined as eQTLs within 4Mb of the transcription start site of the encoding gene. We have added this definition to the methods. We used a nominal p value threshold of 0.05 as a permissive and arbitrary cutoff to compare basic stats of local and distal eQTLs. The eQTLs were a marginal component of the manuscript (only presented in supplemental figures), and changing this threshold did not change the general conclusions about eQTLs that we presented. Namely, local eQTLs outnumber distal eQTLs and local eQTLs tend to be shared across tissues whereas distal eQTLs tend to be tissue specific. This study did not include protein data and we therefore do not report any pQTLs.

(x) Why was heritability computed from called eQTLs rather than from suitably partitioned genetic relationship matrices?

The local heritability of each transcript (regardless of whether it had an eQTL) was calculated using the haplotype probabilities for the marker nearest to the transcription start site of the encoding gene. In typical situations, we would include the kinship matrix as a random effect in this model. However, here we were using the fitted values of the model as the locally encoded transcript level,

and we did not want to confound the model by including distal factors. In reviewing these methods we noticed that we mistakenly said that we used the kinship correction. We have now removed this from the text (Line 145 in the Methods with tracked changes).

(xi) The human validation material is quite interesting but I am not an expert on this part of the paper. It did seem overly long.

Upon second reading of this section, we agree that it could be shortened. We have reduced the length of this section to highlight the most important results (Section starts on Line 337 in manuscript with tracked changes).

(xii) The discussion is incredibly short – there was no attempt to place the findings in a wider context.

We initially included a discussion with a "supplemental discussion" to show that we could reduce the length of the manuscript if needed. We have removed the "supplemental discussion" header to include the full-length discussion, which does discuss the work in a broader context.

Methods:

(i) Genotyping (line 25 onwards) Why were haplotypes determined from RNAseq reads and not from the SNP genotypes?

We apologize for the confusing wording of this section. Haplotypes were determined both from GigaMUGA SNPs and by RNA-Seq. Using both methods provides redundancy as a quality control measure. There were several samples in which the two methods disagreed or had poor quality RNA-Seq data. These mice were excluded from the analysis. We have added text to the methods clarifying why both methods were used (Lines 34-36 in Methods with tracked changes.)

(ii) It not clear whether the CC-RIX mice were kept in the same animal facility as the DO mice or were from a different experiment. Were they on the same high-fat diet as the DO? - please clarify.

To clarify the mouse populations, we have added "CC-RIX" to one of the headers in the methods to indicate that this section describes the CC-RIX mice (Line 40 in Methods with tracked changes). The CC-RIX mice were housed at The Jackson Laboratory. The DO mice were part of a previous experiment and were housed at the University of Wisconsin. DO and CC-RIX mice were maintained on different high-fat, high-sugar diets. DO mice received a HF/HS diet (44.6% kcal fat, 34% carbohydrate, and 17.3% protein) from Envigo Teklad (catalog number

TD.08811). The CC-RIX mice received a custom-designed high-fat, high-sugar (HF/HS) diet (Research Diets D19070208).

(iii) It is not clear what the “processed data” (Methods line 31) refer to. Are these the CC-RIX genotypes? If so, what are the gene expression data?

We agree that this heading is confusing. We have changed the heading to "Pre-processed DO data" and added a sentence to the underlying paragraph that these data were part of a previous publication, and we downloaded them directly from Dryad (Line 38 in Methods with tracked changes).

(iv) TWAS analysis (methods line 250 onwards. Using just the SNP closest to the TSS for each gene might result in underestimating local genetic effects – it would have been better to have taken the most associated SNP within say 100kb. Not all cis SNPs will be associated with the expression trait, so picking one based solely on location is sub-optimal.

We agree that this method would be problematic in human data. However, the mice used in this experiment have large haplotype blocks, and the markers within 100kb of any given marker will have identical or nearly identical genotype distributions across the animals.

Reviewer #3 (Remarks on code availability):

The link to the code <https://figshare.com/DOI:10.6084/m9.figshare.27066979228> does not open

We apologize for this error. We have fixed the URL in the manuscript, and it should now point to the correct page.

Dear Editors and Reviewers,

Thank you again for your careful review of our manuscript and your thoughtful comments. It is very helpful for us to see where important points in the manuscript were still unclear. We have made three main changes to the manuscript based on reviewer comments:

1. We changed Figure 1G to include genetic correlations, phenotype correlations, and heritabilities
2. We added an analysis to assess the role of local eQTL in mediation. These results are now shown in Figure 3B-C and discussed in the main text
3. We removed the eQTL results from the main text to avoid any confusion between eQTL analysis and HDMA.

Our detailed responses to each reviewer comment are below. We have included only the comments that required a response. The original R1 comment and response are included for reference and italicized. The R2 comments are in red, and the R2 responses are indented and in blue. Each new section is also annotated as to whether it is from R1 or R2 and whether it is a reviewer comment or response.

We are including the revised manuscript as well as a version with tracked changes for ease of comparison.

We think that these changes have further enhanced the clarity of the manuscript and will greatly improve reader understanding.

Gregory Carter

Reviewer #3 (Remarks to the Author):

My responses to the rebuttal are in red

R1 review - (i) Is the permutation procedure (Fig 3B,C) valid? It ignores relationships between individuals across omic levels (ie assumes all mice are exchangeable) and so might inflate the apparent significance. If I have understood correctly, the permutation procedure is performed such that it destroys any correlation between the three omic levels – including the existence of any QTLs or eQTLs, and this is likely too harsh a null hypothesis. It might be better to use multivariate generative models to simulate sets of genotypes, transcripts and phenotypes and evaluate performance on those rather than using permutation. An alternative might be to ask how unusual are the correlations $\text{cor}(P_C, T_C)$ etc for the optimal choices of weights compared to randomly sampled weights using the unpermuted data, possibly augmented with a distribution fitted to the random correlations. In any event, that authors should justify their choice of permutation strategy and explain why it supports their thesis.

R1 response - We apologize for the lack of clarity on this critical point. The permutation test was performed by only permuting the transcriptomes. The correlations between genome and phenome were preserved in this test. Thus, we preserved the effects of any QTLs while breaking the association between eQTLs (and all other genetic effect on transcription) with QTLs. The essential goal of this procedure was to determine whether it was possible within a random transcriptome to identify a spurious transcriptomic profile that appeared to mediate the true genotype-phenotype correlations as well as our optimized transcriptomic profile in the true data. Our permutation test definitively rules this possibility out; the true transcriptome is more highly aligned with the genotype-phenotype map than expected at random. We stress that this is nontrivial, given that canonical correlation analysis (CCA)-based approaches have problems overfitting because they involved estimating the inverse of a covariance matrix and, hence, can detect such spurious components. We have added clarification of this point, which can be seen in lines 176-179 in the manuscript with tracked changes. The suggestion to consider the performance of HDMA under generative model hypotheses is an important one, and is the subject of ongoing research, but is beyond the scope of this paper. In this study, we use HDMA in a purely descriptive mode.

We randomly sampled the unpermuted weights 10,000 times and recalculated the correlations $\text{cor}(G_C, T_C)$ and $\text{cor}(T_C, P_C)$ as shown in Figure 3C. The results are shown below. We think it would add more confusion than information to add this to the manuscript figure, but shows clearly that the correlations identified by HDMA are well outside the null distribution. The original permuted values from the manuscript in Figure 3 are shown in gray, and the observed correlations are shown in red.

R2 review - Thanks for clarifying the permutation procedure. So to summarise, the relationship between genotype and phenotypes (ie QTL) is preserved by permutation but there no longer any relationship between transcriptome and either genotype or phenome, (but presumably the internal gene expression network

correlation structure is preserved). Under this permutation procedure we would not expect to see any mediation, therefore. Whilst I understand the rationale for this approach, it is very severe. It does not really address the question of whether the mediation observed (ie the correlation of T_C and P_C) is unusually strong, given the presence of eQTLs; is T_C better correlated with P_C than might be expected given the existence of lots of eQTLs.

At the very minimum, the manuscript should be very clear as to what the permutation procedure is demonstrating and its limitations. Ideally a different approach should be adopted, although I understand that this would be a significant amount of work and quite difficult.

R2 response - We apologize that we have not been able to address this point satisfactorily. To further clarify this procedure, we have added an analysis, the results of which have been added to Figure 3B and C.

We generated two additional transcriptomic kernel matrices to consider the effects of the presence of eQTL in the system. 1) We generated a "local kernel" by imputing gene expression from local haplotype only. This included only the local effects (local eQTL) on transcription. We ran HDMA using each kernel in turn. 2) We also generated a "distal kernel" by first regressing out the effect of local haplotype on each transcript. We used the residuals to generate a kernel matrix that captured only distal effects on transcription.

The path coefficient identified using the local transcriptomic kernel was not significantly different from the null, indicating that when local effects alone are considered, the transcriptome does not mediate the effects of the genome on phenome. In contrast, the path coefficient identified using the distal transcriptomic kernel, was highly significantly different from the null distribution, and indistinguishable from that identified using the full transcriptome.

Looking at the G-T and T-P correlations, we see that the G-T correlation is only slightly lowered by using the distal kernel. It should be a little lower because we have removed some of the genetic effect (local effects) of the genome on the transcriptome. The G-T correlation for the local kernel is extremely high, which also makes sense, because we have defined the transcriptome using the genome. However, the T-P correlation when using the local kernel is very low (0.14), suggesting that animals sharing many local eQTLs are not expected to be very similar in traits.

We have added text to the manuscript and methods to explain this new analysis.

Minor Points:

R1 review - (i) Figure 1 is very good, except Figure 1G could be improved if the upper triangle of the heatmap displayed genetic correlations and the main diagonal heritabilities. This is an interesting way to add more information to the plot. We tried it (shown below), but then realized that it might be confusing to have two different types of data in a single heat map. Because most of the cells show Pearson correlation, but the diagonal has very low values, we think this might cause more confusion, especially since the heritabilities are shown in Figure 1F.

R1 response - This is an interesting way to add more information to the plot. We tried it (shown below), but then realized that it might be confusing to have two different types of data in a single heat map. Because most of the cells show Pearson correlation, but the diagonal has very low values, we think this might cause more confusion, especially since the heritabilities are shown in Figure 1F.

R2 review - The Figure shown above in the response does not correspond to what was requested – I was asking for the upper triangle to be the genetic correlations and the lower triangle the phenotypic correlations, in addition to the main diagonal being the heritabilities. The Figure seems to have just the genetic correlations.

R2 response - We apologize for misunderstanding the initial comment. We think we understand better now what is being asked, and we have revised figure 1G. We now use a tri-color heat map to show the genetic correlations, phenotypic correlations, and heritabilities for the traits. This new panel is pasted below.

R1 review - (ii) Figure 2 could be significantly improved by replacing the violin plots in Fig2A with overlapping distributions as in Figure 1A,B. Fig 2B is informative but I think the use of linear regression is not the best way of showing the shapes of the distributions for Local and Distal are different. Clearly a straight line does not fit any of the data very well. Can the authors think of another measure which quantifies the fact that strong local eQTL are more likely to have small trait correlations than strong distal eQTL?

R1 response - We tried using overlapping distributions (below), but it became difficult to compare the distributions across the tissues. We would like Figure 2A to show a direct comparison of the variance explained by local and distal eQTLs across tissues, so we reverted to the violin plots.

R2 review - This is a matter of personal preference, so I don't insist on the use of overlapping distributions, but I do find the violin plots to be so similar it is hard to draw any conclusions about the distributions. Incidentally the Figure legend now seems to be adjusted for the overlapping distribution plots, not the violin plots.

R2 response - Thank you for catching this error. We have fixed the legend.

R1 review - (iv) Figure 4 D – Please include scatter plots of P_C vs T_C , G_C vs T_C , G_C vs P_C - this is surely key to understanding. Please replace violin plots with overlapping distributions as in Figure 1A,B

R1 response - We tried using overlapping histograms to compare these distributions, but the distributions are so similar that the figure was difficult to read (see below). Instead, we replaced the distributions with separate panels showing the observed loading distribution for each tissue compared with the null. We shaded the area of the distributions that were more extreme than the null distribution and noted the number of genes in each extreme group. This adds a little more information to help distinguish the distributions. The scatter plots took up a lot of room in Figure 3 without adding much information, so we have created a new Supplemental Figure S3 to show the details of these correlations.

R2 review - I think the scatter plots are an excellent addition and should be in the main paper and the violin plots/overlapping distributions in the supplement.

R2 response - We have added the scatter plots to figure 4 and added the overlapping distributions as Supplemental Figure 3.

R1 review - (ix) What were the exact criteria for calling eQTL (local and distal) and pQTL? Were different significance thresholds applied for local vs distal? The methods are vague – why does a LOD score threshold of 8 equate to a p-value of 0.05 (and what does this p-value mean – it is genome wide significance?) Surely it would be better to use an FDR-based threshold.

R1 response - Local eQTLs were defined as eQTLs within 4Mb of the transcription start site of the encoding gene. We have added this definition to the methods. We used a nominal p value threshold of 0.05 as a permissive and arbitrary cutoff to compare basic stats of local and distal eQTLs. The eQTLs were a marginal component of the manuscript (only presented in supplemental figures), and changing this threshold did not change the general conclusions about eQTLs that we presented. Namely, local eQTLs outnumber distal eQTLs and local eQTLs tend to be shared across tissues whereas distal eQTLs tend to be tissue specific. This study did not include protein data and we therefore do not report any pQTLs.

R2 review - I don't think a 5% cutoff is at all reasonable. The manuscript does not even report the threshold or the numbers of QTL in the main text – they are buried in supplemental Figure S2B-E, so I think this point still requires attention. By pQTL I meant physiological QTL, not protein QTL, sorry I should have made that clear.

R2 response - We apologize again for the lack of clarity in our response. We also slightly understated the stringency of the threshold. The LOD score of 8 corresponds to a permutation-based genome-wide threshold < 0.01 . This range threshold is standard practice (PMID 7851788, ref 10 in Methods) and the islet data in this paper have been previously published with a less stringent threshold (PMID 29567659, ref 12).

In light of this comment, we think that we put too much emphasis on the eQTL analysis in this manuscript, and that inclusion of these results will be very confusing to many readers. The purpose of this manuscript is to describe high-dimensional mediation, which is completely independent of QTL mapping of any kind. We performed eQTL mapping only because we thought people would be curious about these basic results. However, in the primary analysis we did not do any mapping. To clarify this point in the manuscript, we have removed the description of the eQTL analysis from the main text and altered the language to better guide the reader through the reasoning of how we used gene expression and local and distal genetic contributions to gene expression.

R1 comment - (ii) It not clear whether the CC-RIX mice were kept in the same animal facility as the DO mice or were from a different experiment. Were they on the same high-fat diet as the DO? - please clarify.

R1 response - To clarify the mouse populations, we have added "CC-RIX" to one of the headers in the methods to indicate that this section describes the CC-RIX mice (Line 40 in Methods with tracked changes). The CC-RIX mice were housed at The Jackson Laboratory. The DO mice were part of a previous experiment and were housed at the University of Wisconsin. DO and CC-RIX mice were maintained on different high-fat, high-sugar diets. DO mice received a HF/HS diet (44.6% kcal fat, 34% carbohydrate, and 17.3% protein) from Envigo Teklad (catalog number TD.08811). The CC-RIX mice received a custom-designed high-fat, high-sugar (HF/HS) diet (Research Diets D19070208).

R1 review - These data should be included in the Methods.

R2 response - These data are included in the methods lines 8-12 and lines 41-51

R1 review - (iv) TWAS analysis (methods line 250 onwards. Using just the SNP closest to the TSS for each gene might result in underestimating local genetic effects – it would have been better to have taken the most associated SNP within say 100kb. Not all cis SNPs will be associated with the expression trait, so picking one based solely on location is sub-optimal.

R1 response - We agree that this method would be problematic in human data. However, the mice used in this experiment have large haplotype blocks, and the markers within 100kb of any given marker will have identical or nearly identical genotype distributions across the animals.

R2 review - I don't agree – it is not true that nearby SNPs will always be surrogates for each other, unless they also share the same strain distribution pattern in the founders.

R2 response - We apologize for a lack of clarity. We did not perform association mapping with SNPs. When working with DO and CC mice, it is standard to perform mapping with ancestral haplotype probabilities. LD blocks of the haplotypes degrade very slowly. As an example, we have pasted below the correlations between markers on Chr 1 at increasing intervals up to 500 kb (bars indicate standard deviation). Even at this distance, the correlation between markers (using haplotypes) is 0.9. The haplotypes closest to the TSS of the gene are an excellent estimate of the haplotypes in the neighborhood. This is one of the reasons we think we see much higher contributions of local genotype to transcript abundance in these mice than is typically seen in humans (discussed in lines 118-130).

Reviewer #3 (Remarks to the Author):

My responses to the rebuttal are in red

This study uses a cohort of diversity outcross mice to perform a novel mediation analysis of obesity-related traits. The two main findings are that (I) distal eQTLs (ie trans eQTLs) seem to play a larger role in controlling transcripts relevant to obesity traits than do local (cis) eQTLs, and (ii) it is possible to define composite measures of genotype and transcriptome and phenotype and perform a mediation analysis on the composite measures, thereby establishing causality.

Both of these findings are interesting and noteworthy and would be of interest to the readership of Nature Comms. However, I also had a large number of queries about the manuscript. Whilst most of these are related to improving the presentation, there are a two more substantive queries which need to be dealt with satisfactorily.

Major points:

(i) Is the permutation procedure (Fig 3B,C) valid? It ignores relationships between individuals across omic levels (ie assumes all mice are exchangeable) and so might inflate the apparent significance. If I have understood correctly, the permutation procedure is performed such that it destroys any correlation between the three omic levels – including the existence of any QTLs or eQTLs, and this is likely too harsh a null hypothesis. It might be better to use multivariate generative models to simulate sets of genotypes, transcripts and phenotypes and evaluate performance on those rather than using permutation. An alternative might be to ask how unusual are the correlations $\text{cor}(P_C, T_C)$ etc for the optimal choices of weights compared to randomly sampled weights using the unpermuted data, possibly augmented with a distribution fitted to the random correlations. In any event, that authors should justify their choice of permutation strategy and explain why it supports their thesis.

We apologize for the lack of clarity on this critical point. The permutation test was performed by only permuting the transcriptomes. The correlations between genome and phenome were preserved in this test. Thus, we preserved the effects of any QTLs while breaking the association between eQTLs (and all other genetic effect on transcription) with QTLs. The essential goal of this procedure was to determine whether it was possible within a random transcriptome to identify a spurious transcriptomic profile that appeared to mediate the true genotype-phenotype correlations as well as our optimized transcriptomic profile in the true data. Our permutation test definitively rules this possibility out; the true transcriptome is more highly aligned with the genotype-phenotype map than expected at random. We stress that this is nontrivial, given that canonical correlation analysis (CCA)-based approaches have problems overfitting because they involved estimating the inverse of a covariance matrix and, hence, can

detect such spurious components. We have added clarification of this point, which can be seen in lines 176-179 in the manuscript with tracked changes.

The suggestion to consider the performance of HDMA under generative model hypotheses is an important one, and is the subject of ongoing research, but is beyond the scope of this paper. In this study, we use HDMA in a purely descriptive mode.

We randomly sampled the unpermuted weights 10,000 times and recalculated the correlations $\text{cor}(G_C, T_C)$ and $\text{cor}(T_C, P_C)$ as shown in Figure 3C. The results are shown below. We think it would add more confusion than information to add this to the manuscript figure, but shows clearly that the correlations identified by HDMA are well outside the null distribution. The original permuted values from the manuscript in Figure 3 are shown in gray, and the observed correlations are shown in red.

Thanks for clarifying the permutation procedure. So to summarise, the relationship between genotype and phenotypes (ie QTL) is preserved by permutation but there no longer any relationship between transcriptome and either genotype or phenome, (but presumably the internal gene expression network correlation structure is preserved). Under this permutation procedure we would not expect to see any mediation, therefore. Whilst I understand the rationale for this approach, it is very severe. It does not really address the question of whether the mediation observed (ie the correlation of T_C and P_C) is unusually strong, given the presence of eQTLs; is T_C better correlated with P_C than might be expected given the existence of lots of eQTLs.

At the very minimum, the manuscript should be very clear as to what the permutation procedure is demonstrating and its limitations. Ideally a different approach should be

adopted, although I understand that this would be a significant amount of work and quite difficult.

(ii) I'm a bit puzzled by the use of the CC-RIX mice as a validation set. It appears that actual body weights were not measured in the CC-RIX (why not? – weight is a standard phenotype) so they were imputed from local transcript data, which complicates their use for validation and makes it far less convincing. I really don't see what they add to the study. Why not simply keep back a random 10% of the DO mice, train the models on the 90% and test the predictions into the 10%, for many random samplings? (ie the standard machine learning cross validation procedure).

We apologize for the lack of clarity on this point. Body weight was measured in the CC-RIX. We used measured body weight as the ground truth validation in the CC-RIX population. We estimated the metabolic disease index (MDI), which is largely based on body weight from the DO and compared it to the actual body weight as the validation (Figure 7B). To estimate MDI, we either used the measured transcriptome in the CC-RIX, or a predicted transcriptome based on local genotype. While the MDI based on the measured transcriptome correlated well with body weight, the MDI based on the locally imputed transcriptome was not correlated with body weight. We interpreted this result as support that genetic effects on genotype are not mediated through local gene regulation. It was important to use the CC-RIX data as opposed to validating within the DO data because the point of local and distal regulation is particularly critical when translating results between populations with different allele structure. We suggest that there is a failure of human TWAS results to translate between human populations because genetic effects are mediated through distal gene regulation, which are dramatically different across populations with different allele structures. In this mouse experiment, the two populations shared ancestral haplotypes, but had dramatically different allele structure, thus allowing us to tease apart the effects of local and distal gene regulation on phenotypic effects.

Thanks for the clarification. I still think there is a case for a training/test set analysis within the DO, in addition to the RIX, because of the challenges interpreting the replication experiments due to differences in diet, husbandry etc between the two populations. That said, it is quite impressive that the RIX replicates the DO despite all these differences and using RIX body weight as a surrogate phenotype.

Minor Points:

(i) Figure 1 is very good, except Figure 1G could be improved if the upper triangle of the heatmap displayed genetic correlations and the main diagonal heritabilities.

This is an interesting way to add more information to the plot. We tried it (shown below), but then realized that it might be confusing to have two different types of data in a single heat map. Because most of the cells show Pearson correlation, but the diagonal has very low values, we think this might cause more confusion, especially since the heritabilities are shown in Figure 1F.

The Figure shown above in the response does not correspond to what was requested – I was asking for the upper triangle to be the genetic correlations and the lower triangle the phenotypic correlations, in addition to the main diagonal being the heritabilities. The Figure seems to have just the genetic correlations.

(ii) Figure 2 could be significantly improved by replacing the violin plots in Fig2A with overlapping distributions as in Figure 1A,B. Fig 2B is informative but I think the use of linear regression is not the best way of showing the shapes of the distributions for Local and Distal are different. Clearly a straight line does not fit any of the data very well. Can the authors think of another measure which quantifies the fact that strong local eQTL are more likely to have small trait correlations than strong distal eQTL?

We tried using overlapping distributions (below), but it became difficult to compare the distributions across the tissues. We would like Figure 2A to show a direct comparison of the variance explained by local and distal eQTLs across tissues, so we reverted to the violin plots.

This is a matter of personal preference, so I don't insist on the use of overlapping distributions, but I do find the violin plots to be so similar it is hard to draw any

conclusions about the distributions. Incidentally the Figure legend now seems to be adjusted for the overlapping distribution plots, not the violin plots.

Figures 2B and C have been changed to avoid the linear test. We binned the transcripts into centiles based on their variance explained. We then calculated the mean and 95th and 5th percentiles of their maximum trait correlation. We smoothed these values using splines. The top and bottom lines in each panel shows the 95th and 5th percentile respectively and the middle line shows the mean. We reported statistics from Spearman rank correlation tests.

This seems OK

(iii) Fig 3 is hard to follow, particularly since it mentions Kernelization which is otherwise not mentioned in the main paper (it's mentioned in the Methods) Not sure what G_K, T_K, P_K signify.

We have added a short description of the kernelization terminology to the main text where we describe high-dimensional mediation. The new text appears in Lines 150-156 of the manuscript with tracked changes.

Thank you

(iv) Figure 4 D – Please include scatter plots of P_C vs T_C, G_C vs T_C, G_C vs P_C - this is surely key to understanding. Please replace violin plots with overlapping distributions as in Figure 1A,B

We tried using overlapping histograms to compare these distributions, but the distributions are so similar that the figure was difficult to read (see below).

Instead, we replaced the distributions with separate panels showing the observed loading distribution for each tissue compared with the null. We shaded the area of the distributions that were more extreme than the null distribution and noted the number of genes in each extreme group. This adds a little more information to help distinguish the distributions.

The scatter plots took up a lot of room in Figure 3 without adding much information, so we have created a new Supplemental Figure S3 to show the details of these correlations.

I think the scatter plots are an excellent addition and should be in the main paper and the violin plots/overlapping distributions in the supplement.

(v) Fig 5 is good. It would be helpful to define exactly what is meant by TWAS in this study. I suggest a completely different color ramp is used to indicate tissue type from that used to indicate heritability – it's a bit confusing. It would also help to report the p-values of the t-tests for comparing the heritability distributions for distal vs local (in the three inset boxplots in the Fig 5)

We have added a short description of our TWAS procedure to the results section that discusses Figure 5. We have also changed the colors for the tissues to select colors that do not overlap with the heat maps or the local/distal color scheme. We added t-test p values to the legend of Figure 5 for each of the box plots (page 14 in manuscript with tracked changes).

Thank you

(vi) Given the diverse ancestry of the DO, involving alleles from three different murine subspecies, it would be interesting to know if the distal or local eQTLs more often involved alleles segregating between subspecies.

This is an interesting question and has been addressed extensively in the literature (PMIDs 25230953, 21406540, 34849860, 34425882, 33264334, 31961859). Because eQTLs were only an incidental piece of this manuscript (presented only in supplemental figures) and because this issue has been examined previously, we performed a limited analysis that is presented only in this response letter. We collected local and distal allele coefficients for all haplotypes. A comparison of the local and distal eQTL coefficients shows that the *castaneus* (green) and *musculus* (red) haplotypes tend to have larger local coefficients than the *domesticus* strains (gray). An ANOVA with Tukey's Honestly Significant Difference showed that the *castaneus*, *musculus*, and the wild-derived *domesticus* strain (WSB) had significantly stronger local eQTL coefficients than the other *domesticus* strains (top figures). For distal coefficients (bottom figures), both *castaneus*, and *musculus* strains had significantly higher coefficients than all *domesticus* strains.

This is a satisfactory response – I think this is an interesting finding, possibly worth adding as a supplemental figure.

(vii) Is there a reason for preferring the nomenclature “Local vs Distal” instead of the more usual “cis vs trans”?

We avoid the usage of cis and trans terminology because these terms have specific biochemical definitions that are not completely captured by physical location on the genome (See Box 2 in PMID: 18597885, ref 22 in the manuscript). Because we are not evaluating the biochemical nature of variants in this study and are classifying variants only by genomic position, we use the terms local and distal. We have added text to the introduction to define these terms and discuss the difference between cis/trans and local/distal terminology.

Although I prefer the use of cis/trans, so long as the paper defines these terms carefully this seems satisfactory

(viii) The statements around line 280: The mean loadings for alpha-cell specific transcripts were significantly greater than 0, while the mean loadings for delta- and endothelial-cell specific genes were significantly less than 0 (Fig. 8B)

The study does not give p-values for these statements – Fig 8B appears to show boxplots which don't give an indication of significance. There are several places in the MS where boxplots are used as evidence of “significance” without a formal p-value being provided.

We apologize for this omission. We have added p values to the text for each of these significance claims.

Thank you

(ix) What were the exact criteria for calling eQTL (local and distal) and pQTL? Were different significance thresholds applied for local vs distal? The methods are vague – why does a LOD score threshold of 8 equate to a p-value of 0.05 (and what does this p-value mean – it is genome wide significance?) Surely it would be better to use an FDR-based threshold.

Local eQTLs were defined as eQTLs within 4Mb of the transcription start site of the encoding gene. We have added this definition to the methods. We used a nominal p value threshold of 0.05 as a permissive and arbitrary cutoff to compare basic stats of local and distal eQTLs. The eQTLs were a marginal component of the manuscript (only presented in supplemental figures), and changing this threshold did not change the general conclusions about eQTLs that we presented. Namely, local eQTLs outnumber distal eQTLs and local eQTLs tend to be shared across tissues whereas distal eQTLs tend to be tissue specific. This study did not include protein data and we therefore do not report any pQTLs.

I don't think a 5% cutoff is at all reasonable. The manuscript does not even report the threshold or the numbers of QTL in the main text – they are buried in supplemental Figure S2B-E, so I think this point still requires attention. By pQTL I meant physiological QTL, not protein QTL, sorry I should have made that clear.

(x) Why was heritability computed from called eQTLs rather than from suitably partitioned genetic relationship matrices?

The local heritability of each transcript (regardless of whether it had an eQTL) was calculated using the haplotype probabilities for the marker nearest to the transcription start site of the encoding gene. In typical situations, we would include the kinship matrix as a random effect in this model. However, here we

were using the fitted values of the model as the locally encoded transcript level, and we did not want to confound the model by including distal factors. In reviewing these methods we noticed that we mistakenly said that we used the kinship correction. We have now removed this from the text (Line 145 in the Methods with tracked changes).

This does not address my question, but it was a fairly minor point, so I don't insist.

(xi) The human validation material is quite interesting but I am not an expert on this part of the paper. It did seem overly long.

Upon second reading of this section, we agree that it could be shortened. We have reduced the length of this section to highlight the most important results (Section starts on Line 337 in manuscript with tracked changes).

Thank you

(xii) The discussion is incredibly short – there was no attempt to place the findings in a wider context.

We initially included a discussion with a "supplemental discussion" to show that we could reduce the length of the manuscript if needed. We have removed the "supplemental discussion" header to include the full-length discussion, which does discuss the work in a broader context.

Thank you

Methods:

(i) Genotyping (line 25 onwards) Why were haplotypes determined from RNAseq reads and not from the SNP genotypes?

We apologize for the confusing wording of this section. Haplotypes were determined both from GigaMUGA SNPs and by RNA-Seq. Using both methods provides redundancy as a quality control measure. There were several samples in which the two methods disagreed or had poor quality RNA-Seq data. These mice were excluded from the analysis. We have added text to the methods clarifying why both methods were used (Lines 34-36 in Methods with tracked changes.)

Thank you

(ii) It not clear whether the CC-RIX mice were kept in the same animal facility as the DO mice or were from a different experiment. Were they on the same high-fat diet as the DO? - please clarify.

To clarify the mouse populations, we have added "CC-RIX" to one of the headers in the methods to indicate that this section describes the CC-RIX mice (Line 40 in Methods with tracked changes). The CC-RIX mice were housed at The Jackson Laboratory. The DO mice were part of a previous experiment and were housed at the University of Wisconsin. DO and CC-RIX mice were maintained on different high-fat, high-sugar diets. DO mice received a HF/HS diet (44.6% kcal fat, 34% carbohydrate, and 17.3% protein) from Envigo Teklad (catalog number TD.08811). The CC-RIX mice received a custom-designed high-fat, high-sugar (HF/HS) diet (Research Diets D19070208).

These data should be included in the Methods.

(iii) It is not clear what the "processed data" (Methods line 31) refer to. Are these the CC-RIX genotypes? If so, what are the gene expression data?

We agree that this heading is confusing. We have changed the heading to "Pre-processed DO data" and added a sentence to the underlying paragraph that these data were part of a previous publication, and we downloaded them directly from Dryad (Line 38 in Methods with tracked changes).

Thank you

(iv) TWAS analysis (methods line 250 onwards. Using just the SNP closest to the TSS for each gene might result in underestimating local genetic effects – it would have been better to have taken the most associated SNP within say 100kb. Not all cis SNPs will be associated with the expression trait, so picking one based solely on location is sub-optimal.

We agree that this method would be problematic in human data. However, the mice used in this experiment have large haplotype blocks, and the markers within 100kb of any given marker will have identical or nearly identical genotype distributions across the animals.

I don't agree – it is not true that nearby SNPs will always be surrogates for each other, unless they also share the same strain distribution pattern in the founders.

Reviewer #3 (Remarks on code availability):

The link to the code <https://figshare.com/DOI:10.6084/m9.figshare.27066979228> does not open

We apologize for this error. We have fixed the URL in the manuscript, and it should now point to the correct page.

Thank you